# PAS: Estimating the target accuracy before domain adaptation

**Raphaella Diniz, Jackson de Faria Júnior & Martin Ester**
School of Computing Science
Simon Fraser University
Canada
{raphaella_diniz, jackson_de_faria_junior, ester}@sfu.ca

## ABSTRACT

The goal of domain adaptation is to make predictions for unlabeled samples from a target domain with the help of labeled samples from a different but related source domain. The performance of domain adaptation methods is highly influenced by the choice of source domain and pre-trained feature extractor. However, the selection of source data and pre-trained model is not trivial due to the absence of a labeled validation set for the target domain and the large number of available pre-trained models. In this work, we propose **Potential Adaptability Score (PAS)**, a novel score designed to estimate the transferability of a source domain set and a pre-trained feature extractor to a target classification task before actually performing domain adaptation. **PAS** leverages the generalization power of pre-trained models and assesses source-target compatibility based on the pre-trained feature embeddings. We integrate **PAS** into a framework that indicates the most relevant pre-trained model and source domain among multiple candidates, thus improving target accuracy while reducing the computational overhead. Extensive experiments on image classification benchmarks demonstrate that **PAS** correlates strongly with actual target accuracy and consistently guides the selection of the best-performing pre-trained model and source domain for adaptation.

## 1 INTRODUCTION

In many real applications, data is collected from diverse domains, e.g., data obtained from different equipment, collecting procedures, geographic locations, or periods in time. Such differences may lead to a distribution shift between the domains that must be assessed. Unsupervised domain adaptation is a paradigm where only unlabeled data is available for the domain of interest, the target domain. However, labeled data is obtained from a related source domain.

One factor that affects the success of domain adaptation methods is the choice of the source domain data. Domain adaptation methods often rely on many assumptions about the relationship between source and target domains, like the existence of invariant discriminative features, the similarity of the label distribution, or the invariance of the task. Unfortunately, as the labels for the target samples are not available, such assumptions may not be verified in real applications for selecting the most appropriate source data. Violating the data assumptions and

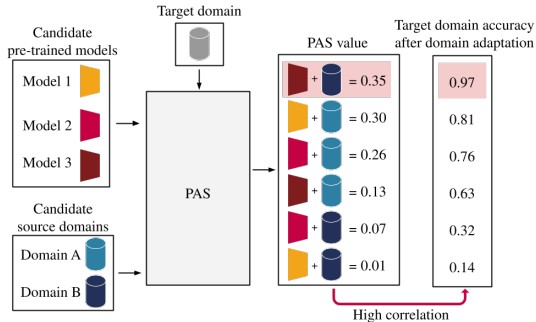

Figure 1: The **Potential Adaptability Score (PAS)** estimates the performance of adapting to an unlabeled target domain given a pre-trained feature extractor and a labeled source domain. It helps in the selection of the best pre-trained model and best source domain among many candidates and is highly correlated with the final target accuracy after domain adaptation.

considering an irrelevant or distant source domain may introduce noise and conflicting patterns during the domain adaptation process. In the worst-case scenario, selecting an undesirable source domain may hurt the target domain performance, a scenario known as negative transfer Zhang et al. (2022). If many source domains are available, it is reasonable to assume that not all of them may contribute equally to the target adaptation. Wisely selecting the source domain that may improve the performance on the target data while avoiding negative transfer is an essential requirement in many real-world applications.

Another key factor that influences the domain adaptation performance is the choice of the pre-trained model. Pre-training on large-scale data allows the models to learn generic features and patterns that are often transferable across domains and tasks, making them valuable for domain adaptation. Recently, practitioners can choose from a vast number of publicly available pre-trained models, spanning diverse architectures and training paradigms. Each pre-trained model may have its own inductive bias and may capture distinct patterns in the data that may be more or less useful when transferring knowledge between domains.

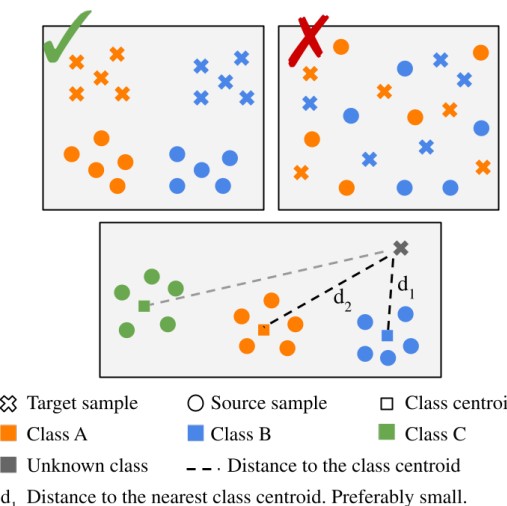

Target sample ⬡   Source sample ○   Class centroid □

■ Class A    ■ Class B    ■ Class C

■ Unknown class    − − · Distance to the class centroid

$d_1$ Distance to the nearest class centroid. Preferably small.

$d_2$ Distance to the second nearest class centroid. Preferably large.

Figure 2: Source and target samples in the embedding space of a pre-trained model. *(top left)* Ideally, a target sample from a given class should be more similar, and hence closer in the embedding space of the pre-trained model, to a source sample from the same class. *(top right)* If new discriminative features need to be learned, the chances of overfitting on the source domain during adaptation increase. *(bottom)* Illustration of the distances from a target sample to all source class centroids. Our **PAS** score considers the relationship between distances $d_1$ and $d_2$, which correspond to the shortest and second shortest distances, respectively.

Despite the importance of selecting a suitable source data and a pre-trained model for the success of domain adaptation, it is still an underexplored topic. Current methods of transferability estimation aim to select the best pre-trained model for transfer learning. However, these methods are not applicable to the domain adaptation scenario since they require target labels Bao et al. (2019); Nguyen et al. (2020); You et al. (2021). One could employ such methods for selecting the best pre-training model using only the labeled source data and ignoring the unlabeled target data. Nevertheless, considering the target data is essential, as transferring to an easy target domain should lead to different results than transferring to a harder one.

Another approach for the problem would be performing domain adaptation for each combination of available source domains and pre-trained models, and applying some model selection strategy Ericsson et al. (2023); You et al. (2019); Sun et al. (2021). However, this approach is very time-consuming since it needs to run a domain adaptation algorithm for each combination.

A third approach would be to measure the distance between source and target feature distributions in the embedding space of the pre-trained model. This approach is also challenging as the popular metrics for the distance between two feature distributions are symmetric, e.g., Maximum Mean Discrepancy (MMD) Gretton et al. (2006), Wasserstein distance Vallender (1974), and CORAL Sun & Saenko (2016). However, a metric suitable for our scenario should be asymmetric because transferring from an easy to a hard domain is more challenging than transferring from the harder domain to the easier one.

In this work, we examine the interplay between the three key components in the domain adaptation setting for classification: (1) target data, (2) source data, and (3) the pre-trained model. We propose the **Potential Adaptability Score (PAS)**, a simple but effective novel measure to quantify the potential success of using a pre-trained model to transfer knowledge from a source domain to the target

domain. Our experiments show how the **PAS** score is highly correlated to the final target accuracy after adaptation.

To the best of our knowledge, this is the first proposal for transferability estimation for the domain adaptation setting. We demonstrate how **PAS** can help to select the most relevant source domain and/or pre-trained model among a set of candidates, indicating the options that are most likely to lead to the best accuracy on the unlabeled target data (See an overview in figure 1.). Our framework selects the most suitable options before actually performing domain adaptation, demanding fewer computational resources and reducing the training time.

**PAS** leverages the generalization power of models pre-trained on a large-scale dataset, such as the popular ImageNet-1k Deng et al. (2009). Specifically for domain adaptation, initializing with a good pre-trained model appears to be a fundamental step in achieving a good transferability between domains Peng et al. (2018); Tang & Jia (2023); Kim et al. (2022); Li et al. (2023); Teterwak et al. (2023). We assume that a good pre-trained model can extract general discriminative features that are robust across all domains. If this assumption is true, samples from the same class are expected to be closer together in the embedding space generated by the pre-trained model, compared to samples from different classes, even in the presence of feature distribution shift. This ideal scenario is illustrated in the top left of the figure 2. Otherwise, as shown in the example on the top right of the figure, the model should learn new discriminative features during the adaptation from the limited labeled source data to enable the classification task, increasing the chances of overfitting to the source domain. Our **PAS** score is inspired by the Silhouette score, used for assessing the consistency of data clusters Rousseeuw (1987). We modify the original Silhouette score to measure the similarity of the unlabeled target samples to some of the known source class clusters defined in the pre-trained embedding space.

We summarize our contributions as follows:

- We propose **PAS**, a simple novel measure to quantify the potential contribution of a pre-trained model and labeled source domain in the adaptation to an unlabeled target domain before performing domain adaptation.
- We propose a framework to select the most relevant pre-trained model or source domain from a collection of potential candidates for performing domain adaptation.
- We empirically validate our framework using different domain adaptation methods and image classification benchmarks, and show how our score has a high correlation with the target accuracy.

## 2 RELATED WORK

**Unsupervised domain adaptation.** The goal of unsupervised domain adaptation (UDA) is to transfer the knowledge learned from a labeled domain to a different unlabeled target domain. Usually, this goal is achieved by learning a latent representation that is invariant across domains. Several works minimize the distribution discrepancy on the representation using statically defined distance metrics such as Maximum Mean Discrepancy (MMD) (e.g., DAN Long et al. (2015), DDC Tzeng et al. (2014), JAN Long et al. (2017)), covariance (e.g., DCORAL Sun & Saenko (2016)), or Wasserstein distance (e.g., DeepJDOT Damodaran et al. (2018)). The popularization of generative models inspired the proposal of methods that adopt adversarial learning to align data across different domains. DANN Ganin et al. (2016), CDAN Long et al. (2018), and ADDA Tzeng et al. (2017) are examples of widely adopted UDA methods that have shown promising results. Self-training is another promising paradigm that exploits the pseudo-labels predicted for the target domain to enhance the model. CST Liu et al. (2021), CRST Zou et al. (2019), FixMatch Sohn et al. (2020) and MCC Jin et al. (2020) are examples of methods that explore pseudo-labeling. Most recently, with the dissemination of transformers and foundation models, new works explore the cross-attention mechanism to propose transformer-based domain adaptation methods, such as PMTrans Zhu et al. (2023) and DoT Ren et al. (2024). See Liu et al. (2022); Deng & Jia (2023); Alijani et al. (2024) for a comprehensive survey on domain adaptation methods.

**Pre-training and domain adaptation** Recent works suggest that the choice of the pre-trained feature extractor can significantly improve the result of domain adaptation methods. Teterwak et al. (2023) show that simply adopting a model with better weight initialization can help the robustness

of a model to out-of-distribution samples. Similarly, Kim et al. (2022) empirically show that SOTA pre-training outperforms SOTA domain adaptation methods even without access to a target domain. With a modern pre-trained backbone, older domain adaptation methods perform better than SOTA methods, but no method is consistently better in all benchmarks, and negative transfer can occur. Li et al. (2023) empirically show how, in some cases, the performance of the pre-trained model in an unseen target domain is already decent. However, no single pre-trained model performs well in all target datasets. Tang & Jia (2023) study the effects of pre-training on the domain adaptation between synthetic and real images. Without pre-training, none of the methods considered in the study outperformed the baseline trained only on the labeled source data. Other studies have also proposed new datasets and pre-training techniques that achieve competitive performance in the target domain Shen et al. (2022); Luo et al. (2024). We leverage the potential relationship between pre-training and domain adaptation success to estimate transferability between domains.

**Transferability estimation**    In the past years, many works have proposed scorees for quantitatively estimating the transferability of a pre-trained model to a target task. One of the primary practical applications of such estimation is selecting the best pre-trained model for fine-tuning on the target data. H-score Bao et al. (2019), NCE Tran et al. (2019), LEEP Nguyen et al. (2020) and LogME You et al. (2021) are widely adopted transferability estimation scores. More closely related to our proposal, some works propose scores for transferability estimation by examining the separability of classes in the embedding space encoded by the pre-trained model. Pándy et al. (2022) apply the Bhattacharyya coefficient to quantify the target class separability. Similarly, Meiseles & Rokach (2020) employ the Silhouette score to assess the transferability of time series data. The current methods on transferability estimation focus on the transfer learning problem, where a pre-trained model is adapted to a target task with a few labeled samples. Unfortunately, these methods can not be applied to the domain adaptation problem, where the target labels are not available.

## 3 METHOD

### 3.1 DEFINITIONS

Unsupervised domain adaptation aims to transfer knowledge from a labeled source domain to an unlabeled target domain in the presence of distribution shift. Let $\mathcal{X} \subseteq \mathbb{R}^d$ define the input space and $\mathcal{Y} = \{1, \ldots, C\}$ the label space. The labeled source dataset is denoted by $\mathcal{D}^S = \{(x_i^S, y_i^S)\}_{i=1}^{|\mathcal{D}^S|}$ and the unlabeled target dataset is denoted by $\mathcal{D}^T = \{x_i^T\}_{j=1}^{|\mathcal{D}^T|}$, with $x_i^S, x_i^T \in \mathcal{X}$ and $y_i^S \in \mathcal{Y}$. $S_c^S$ denotes the set of source samples from class $c$. The source and target feature distributions are sampled from different but related distributions, $P_S(\mathcal{X})$ and $P_T(\mathcal{X})$, respectively, being $P_S \neq P_T$. This scenario is also known as *covariate shift*. The goal is to learn a hypothesis $h : \mathcal{X} \to \mathcal{Y}$ that performs well on the target domain.

Let $\theta$ be the parameters of a feature extractor $f_\theta : \mathcal{X} \to \mathcal{Z}$ pre-trained on a large-scale dataset. $z_i^S = f_\theta(x_i^S)$ and $z_i^T = f_\theta(x_i^T)$ denote, respectively, the embedding of a source and a target sample in the embedding space defined by $f_\theta$.

### 3.2 ASSUMPTIONS

We assume that a good pre-trained model $f_\theta$ is able to extract a wide range of patterns and high-level concepts from an input, including discriminative features that are invariant across different domains. We expect that samples from the same class are more similar, having many concepts in common. As a result, two samples from the same class should be closer together in the embedding space $\mathcal{Z}$, no matter the domain. On the other hand, samples from different classes should have very few concepts in common, resulting in a dissimilar embedding representation. Due to the distribution shift between the source and target domains, samples from the same domain are expected to have more concepts in common and, therefore, have more similar representations than samples from different domains. Such assumptions lead to a scenario similar to the one represented in the top left of figure 2. The embeddings of samples from the same domain and class form a well-defined cluster in the space encoded by $f_\theta$. Also, the clusters of samples from the same class, but different domains, are closer together and, ideally, both are distant from all the other clusters.

To summarize, we assume that 1) a good pre-trained model can extract invariant discriminative features, 2) samples from the same class are close in the embedding space, even if they are from different domains, and 3) samples from different classes are distant in the embedding space. Similar assumptions are proposed by Shen et al. (2022) when studying the generalization of embeddings to out-of-distribution samples.

### 3.3 THE POTENTIAL ADAPTABILITY SCORE

We introduce the Potential Adaptability Score (**PAS**) as a measure of the distance from a labeled source dataset to an unlabeled target dataset in the embedding space encoded by a pre-trained feature extractor. The **PAS** score is based on the expectation that each target sample is as close as possible to source samples from a single class and significantly distant from source samples from all other classes in the embedding space defined by a pre-trained model $f_\theta$. This means that a target sample is very similar to source samples from one class and has only a few concepts in common with source samples from all other classes, as illustrated in figure 2. The higher the **PAS** value, the stronger the evidence that the pre-trained model can identify invariant discriminative features between the domains and, consequently, the higher the chances that the pre-trained feature extractor $f_\theta$ has a good transferability from the source to the target samples.

The samples are normalized to unit length, and the distance between samples is calculated using the cosine distance. We assume that the samples from the class $c \in \mathcal{Y}$ are clustered together. We follow Dhillon & Modha (2001) and compute the centroid of each source class cluster $c$ so they represent the vector that, on average, has the highest cosine similarity to all the samples in the cluster.

$$\mu_c = \frac{\sum_{x_i^S \in S_c^S} f_\theta(x_i^S)}{\| \sum_{x_i^S \in S_c^S} f_\theta(x_i^S)\|}. \tag{1}$$

For each target sample $x_i^T$, we calculate its cosine distance to the centroid of each source cluster:

$$\text{dist}(f_\theta(x_i^T), \mu_c) = 1 - (f_\theta(x_i^T) \cdot \mu_c). \tag{2}$$

Let $D_i = \{\text{dist}(f_\theta(x_i^T), \mu_1), ..., \text{dist}(f_\theta(x_i^T), \mu_C)\}$ be the set of distances of the $j$-th target sample to all the source clusters and $sort(D_i)$ the sorted version of the set in ascending order. We define $d_{1i} = sort(D_i)[1]$ and $d_{2i} = sort(D_i)[2]$ as the shortest and the second shortest of the distances, respectively, as illustrated at the bottom of figure 2.

Finally, the **PAS** score is defined by

$$\mathbf{PAS}(\theta, \mathcal{D}^\mathbf{S}, \mathcal{D}^\mathbf{T}) = \frac{1}{|\mathcal{D}^T|} \sum_i^{|\mathcal{D}^T|} \frac{d_{2i} - d_{1i}}{d_{2i}}. \tag{3}$$

Given one or more candidate source domains and a set of pre-trained models, the **PAS** score can help to select the options that are more likely to lead to the best accuracy on the target samples. The selection is done by computing the **PAS** score for each trio of target domain, source domain, and pre-trained model. The combination with the highest **PAS** value is chosen. The selection is done before any domain adaptation training. A single-source domain adaptation method can then be trained with the selected source domain and pre-trained feature extractor.

Our **PAS** score is inspired by the Silhouette score, used for assessing the consistency of data clusters Rousseeuw (1987). The Silhouette is a supervised score calculated by $(b - a)/max\{a, b\}$, where $a$ is the mean intra-cluster distance and $b$ is the mean nearest-cluster distance for each sample. It ranges from $-1$ to $1$, with higher values indicating strong intra-class cohesion and clear inter-class separation. Note that the Silhouette score is fully supervised and designed for IID samples and its original form is not suitable for the domain adaptation problem. Our **PAS** score is an adaptation to accommodate unlabeled target samples and domain shift. We consider the closest source cluster as the true class for each target sample. This assumption makes $a$ always smaller than $b$, and restricts our score to the interval $[0, 1]$. The **PAS** score is close to one if the samples from the target domain

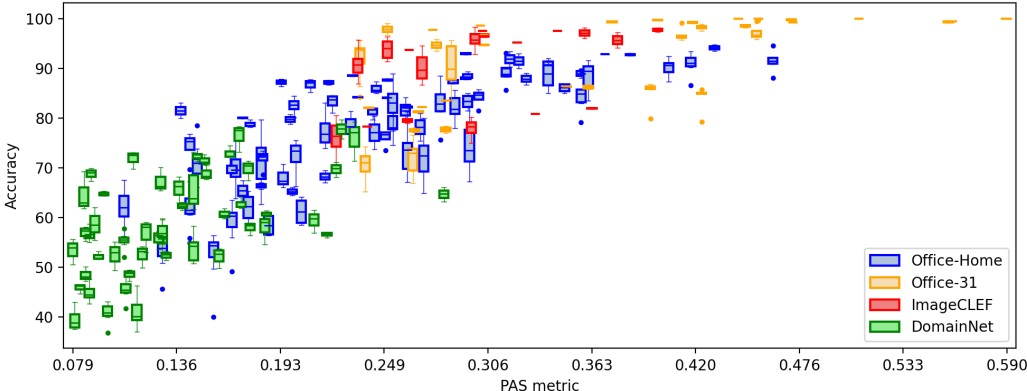

Figure 3: The correlation between the **PAS** score value and the target accuracy after the domain adaptation. Each box summarizes the target accuracy of different domain adaptation methods for a given source-target pair and a pre-trained feature extractor. Higher values for the **PAS** score are strongly correlated with higher target accuracy.

are similar to the centroid of the source class cluster. However, due to the mismatch between the domains, the target samples exhibit a shift in the feature distribution, making $a$ larger than in the IID scenario. As a result, the values for our score are typically smaller. Alternative design choices are discussed and evaluated in section 4.4.

## 4 EXPERIMENTS

**Datasets.** We evaluate **PAS** on four of the most popular benchmarks for domain adaptation: **Office-Home** Venkateswara et al. (2017), **Office-31** Saenko et al. (2010), **ImageCLEF** [1], and **DomainNet** Peng et al. (2019). The benchmarks' statistics are listed in the table 2.

**Domain adaptation methods** Many domain adaptation methods have been proposed in the literature, but none have consistently outperformed the others in all benchmarks. For this reason, we obtained the published target accuracies of a large variety of state-of-the-art methods. We consider methods based on different paradigms and trained using diverse pre-trained feature extractors. The accuracy values are obtained from the popular open-source *Tllib* library for transfer learning Jiang et al. (2022);

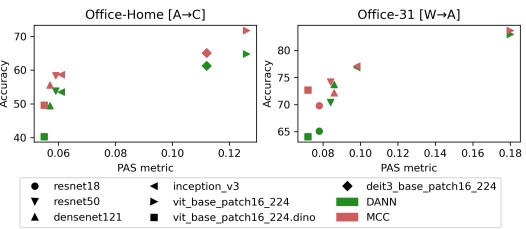

Figure 4: The **PAS** value and target accuracy for the *DANN* and *MCC* methods using different pretrained feature extractors. The **PAS** score can help to select the feature extractor that leads to higher accuracy. *(left)* A→C adaptation in the *Office-Home* benchmark. *(right)* W→A adaptation in the *Office-31* benchmark.

Junguang Jiang (2020), Wang et al. (2023), and from the original papers.

**Baselines** To the best of our knowledge, **PAS** is the first *asymmetric* score proposed for transferability estimation for domain adaptation. We, therefore, compare **PAS** with the symetric metrics **Maximum Mean Discrepancy (MMD)** Gretton et al. (2012) and $\mathcal{A}$-**distance** Peng et al. (2019). The MMD distance is computed using a Gaussian kernel. Due to the quadratic nature of MMD, we restrict its computation to a maximum of 10,000 randomly selected samples per domain for the DomainNet benchmark. The $\mathcal{A}$-**distance** is computed using C-Support Vector Classification. We also report the results for an oracle baseline. The oracle is similar to **PAS**, defined as $\frac{1}{|\mathcal{D}^T|}\sum_i^{|\mathcal{D}^T|}\frac{d_{2i}-d_{1i}}{max\{d_{1i},d_{2i}\}}$. The $d_{1i}$ distance is the cosine distance to the centroid of the true class of the target sample (not known in real scenarios), and $d_{2i}$ is the distance to the closest cluster's

[1]http://imageclef.org/2014/adaptation

Table 1: Average target accuracy of domain adaptation methods and transferability scores for different image classification benchmarks. The highest values are highlighted. Our **PAS** has a high correlation with the target accuracy and, for each target domain, attributes the highest value for the source domain that leads to the highest target accuracy in most scenarios. * Oracle baseline that considers the target labels.

### (a) Office-Home

| Model | Method | A·C | A·P | A·R | C·A | C·P | C·R | P·A | P·C | P·R | R·A | R·C | R·P | Pearson | Spearman |
|---|---|---|---|---|---|---|---|---|---|---|---|---|---|---|---|
| ResNet-50 | Acc. (avg.) | 62.0 | 61.7 | **72.5** | 53.5 | 52.4 | **58.9** | 71.0 | 70.0 | **82.1** | 77.2 | 70.8 | **78.7** | | |
| | MMD (neg.) | -0.135 | -0.113 | **-0.052** | -0.135 | **-0.097** | -0.125 | -0.113 | -0.097 | **-0.033** | -0.052 | -0.125 | **-0.033** | 0.77 | 0.72 |
| | A-distance (neg.) | -1.876 | -1.810 | **-1.333** | -1.876 | -1.827 | **-1.814** | -1.810 | -1.827 | **-1.424** | **-1.333** | -1.814 | -1.424 | 0.76 | 0.78 |
| | **PAS (our)** | 0.107 | 0.143 | **0.201** | 0.128 | 0.156 | **0.166** | 0.182 | 0.168 | **0.288** | 0.217 | 0.147 | **0.254** | 0.81 | 0.82 |
| | Oracle* | 0.041 | 0.037 | 0.093 | -0.022 | -0.018 | -0.022 | 0.103 | 0.100 | 0.218 | 0.165 | 0.096 | 0.195 | 0.98 | 0.93 |
| DeiT-Small | Acc. (avg.) | 74.3 | 71.7 | **76.0** | **61.8** | 58.5 | 61.2 | 81.7 | 83.2 | **86.0** | 83.3 | 82.5 | **84.1** | | |
| | MMD (neg.) | -0.106 | -0.058 | **-0.024** | -0.106 | **-0.077** | -0.102 | -0.058 | -0.077 | **-0.026** | **-0.024** | -0.102 | -0.026 | 0.56 | 0.57 |
| | A-distance (neg.) | -1.865 | -1.761 | **-1.26** | -1.865 | -1.843 | **-1.796** | -1.761 | -1.843 | **-1.37** | **-1.26** | -1.796 | -1.37 | 0.52 | 0.57 |
| | **PAS (our)** | 0.143 | 0.183 | **0.25** | 0.175 | 0.186 | **0.204** | 0.261 | 0.221 | **0.348** | 0.295 | 0.2 | **0.301** | 0.67 | 0.78 |
| | Oracle* | 0.086 | 0.112 | 0.18 | 0.038 | 0.037 | 0.047 | 0.194 | 0.155 | 0.291 | 0.243 | 0.147 | 0.246 | 0.90 | 0.93 |
| DeiT-Base | Acc. (avg.) | **81.5** | 78.8 | 81.1 | **69.9** | 65.4 | 68.0 | 86.0 | 86.7 | **90.6** | 87.4 | 87.2 | **88.5** | | |
| | MMD (neg.) | -0.099 | -0.056 | **-0.028** | -0.099 | **-0.091** | -0.11 | -0.056 | -0.091 | **-0.023** | **-0.028** | -0.11 | -0.023 | 0.56 | 0.57 |
| | A-distance (neg.) | -1.832 | -1.81 | **-1.372** | -1.832 | -1.907 | **-1.823** | -1.81 | -1.907 | **-1.502** | **-1.372** | -1.823 | -1.502 | 0.48 | 0.51 |
| | **PAS (our)** | 0.138 | 0.176 | **0.243** | 0.166 | 0.172 | **0.194** | 0.245 | 0.209 | **0.339** | 0.287 | 0.193 | **0.295** | 0.65 | 0.73 |
| | Oracle* | 0.09 | 0.112 | 0.184 | 0.048 | 0.049 | 0.062 | 0.191 | 0.158 | 0.293 | 0.245 | 0.154 | 0.248 | 0.88 | 0.88 |
| ViT-Small | Acc. (avg.) | 80.1 | 79.8 | **82.2** | 66.4 | 65.2 | **68.2** | 84.1 | 84.2 | **89.0** | 88.0 | 87.2 | **88.5** | | |
| | MMD (neg.) | -0.116 | -0.082 | **-0.032** | -0.116 | **-0.115** | -0.122 | -0.082 | -0.115 | **-0.034** | **-0.032** | -0.122 | -0.034 | 0.61 | 0.49 |
| | A-distance (neg.) | -1.885 | -1.883 | **-1.348** | -1.885 | -1.927 | **-1.862** | -1.883 | -1.927 | **-1.515** | **-1.348** | -1.862 | -1.515 | 0.53 | 0.59 |
| | **PAS (our)** | 0.172 | 0.198 | **0.262** | 0.182 | 0.199 | **0.217** | 0.251 | 0.235 | **0.357** | 0.294 | 0.219 | **0.316** | 0.68 | 0.83 |
| | Oracle* | 0.132 | 0.147 | 0.212 | 0.084 | 0.102 | 0.113 | 0.211 | 0.195 | 0.321 | 0.26 | 0.189 | 0.285 | 0.87 | 0.92 |
| ViT-Base | Acc. (avg.) | 83.0 | 82.7 | **84.5** | 73.2 | 72.2 | **74.4** | 88.3 | 88.6 | **91.4** | 90.2 | 89.5 | **90.8** | | |
| | MMD (neg.) | -0.101 | -0.069 | **-0.031** | **-0.101** | -0.106 | -0.11 | -0.069 | -0.106 | **-0.025** | **-0.031** | -0.11 | -0.025 | 0.59 | 0.57 |
| | A-distance (neg.) | -1.85 | -1.845 | **-1.389** | **-1.85** | -1.952 | -1.885 | -1.845 | -1.952 | **-1.595** | **-1.389** | -1.885 | -1.595 | 0.47 | 0.51 |
| | **PAS (our)** | 0.254 | 0.28 | **0.357** | 0.262 | 0.271 | **0.296** | 0.361 | 0.339 | **0.462** | 0.405 | 0.316 | **0.417** | 0.76 | 0.85 |
| | Oracle* | 0.215 | 0.233 | 0.311 | 0.173 | 0.188 | 0.207 | 0.317 | 0.295 | 0.425 | 0.37 | 0.286 | 0.382 | 0.88 | 0.92 |
| Swin-Base | Acc. (avg.) | **88.5** | 87.7 | 87.9 | **78.3** | 77.1 | 78.2 | 91.5 | 91.9 | **94.2** | 92.9 | **93.0** | 92.8 | | |
| | MMD (neg.) | -0.081 | -0.085 | **-0.039** | **-0.081** | -0.104 | -0.097 | -0.085 | -0.104 | **-0.033** | -0.039 | -0.097 | **-0.033** | 0.48 | 0.45 |
| | A-distance (neg.) | -1.853 | -1.892 | **-1.401** | **-1.853** | -1.95 | -1.917 | -1.892 | -1.95 | **-1.57** | **-1.401** | -1.917 | -1.57 | 0.42 | 0.37 |
| | **PAS (our)** | 0.232 | 0.251 | **0.327** | 0.231 | 0.244 | **0.269** | 0.323 | 0.318 | **0.43** | 0.37 | 0.294 | **0.384** | 0.72 | 0.72 |
| | Oracle* | 0.198 | 0.214 | 0.287 | 0.162 | 0.177 | 0.195 | 0.295 | 0.282 | 0.403 | 0.343 | 0.27 | 0.356 | 0.83 | 0.81 |

### (b) Office-31

| Model | Method | A·D | A·W | D·A | D·W | W·A | W·D | Pearson | Spearman |
|---|---|---|---|---|---|---|---|---|---|
| ResNet-50 | Acc. (avg.) | **71.8** | 70.6 | 90.5 | **100.0** | 91.9 | **98.3** | 0.71 | 0.72 |
| | MMD (neg.) | **-0.145** | -0.165 | -0.145 | **-0.046** | -0.165 | **-0.046** | 0.72 | **0.83** |
| | A-distance (neg.) | -2.00 | -2.00 | -2.00 | -1.783 | -2.00 | -1.783 | 0.72 | **0.83** |
| | **PAS (our)** | **0.265** | 0.239 | 0.246 | **0.423** | 0.188 | **0.407** | **0.73** | 0.66 |
| | Oracle* | 0.192 | 0.166 | 0.246 | 0.445 | 0.188 | 0.407 | 0.80 | 0.83 |
| DeiT-Small | Acc. (avg.) | **77.7** | 77.6 | 94.7 | **99.8** | 94.65 | **98.5** | | |
| | MMD (neg.) | **-0.123** | -0.129 | -0.123 | **-0.058** | -0.129 | **-0.058** | 0.66 | 0.84 |
| | A-distance (neg.) | -2.00 | -1.994 | -2.00 | -1.969 | -1.994 | -1.969 | 0.65 | 0.60 |
| | **PAS (our)** | **0.283** | 0.266 | 0.304 | **0.472** | 0.278 | **0.447** | **0.72** | **0.94** |
| | Oracle* | 0.2 | 0.193 | 0.263 | 0.465 | 0.239 | 0.438 | 0.80 | 1.00 |
| DeiT-Base | Acc. (avg.) | 81.3 | **82.0** | 96.8 | **100.0** | 97.9 | **99.2** | | |
| | MMD (neg.) | **-0.113** | -0.134 | -0.113 | **-0.074** | -0.134 | **-0.074** | 0.54 | 0.60 |
| | A-distance (neg.) | -2.00 | -2.00 | -2.00 | -2.00 | -2.00 | -2.00 | 0.0 | 0.0 |
| | **PAS (our)** | **0.268** | 0.241 | 0.304 | **0.443** | 0.251 | **0.418** | 0.66 | 0.71 |
| | Oracle* | 0.212 | 0.192 | 0.273 | 0.44 | 0.224 | 0.414 | 0.73 | 0.89 |
| ViT-Small | Acc. (avg.) | **83.5** | 82.2 | 98.6 | **100.0** | 97.7 | **99.2** | | |
| | MMD (neg.) | **-0.175** | -0.197 | -0.175 | **-0.098** | -0.197 | **-0.098** | 0.56 | 0.84 |
| | A-distance (neg.) | -2.00 | -2.00 | -2.00 | -2.00 | -2.00 | -2.00 | 0.0 | 0.0 |
| | **PAS (our)** | **0.283** | 0.27 | 0.302 | **0.509** | 0.276 | **0.473** | 0.61 | **0.94** |
| | Oracle* | 0.23 | 0.22 | 0.286 | 0.506 | 0.256 | 0.467 | 0.69 | 1.00 |
| ViT-Base | Acc. (avg.) | 84.0 | **85.0** | 97.2 | **100.0** | 96.8 | **99.3** | | |
| | MMD (neg.) | **-0.098** | -0.118 | -0.098 | **-0.071** | -0.118 | **-0.071** | 0.57 | 0.72 |
| | A-distance (neg.) | -2.00 | -2.00 | -2.00 | -1.953 | -2.00 | -1.953 | 0.64 | 0.72 |
| | **PAS (our)** | **0.423** | 0.395 | 0.453 | **0.59** | 0.412 | **0.558** | 0.71 | 0.83 |
| | Oracle* | 0.373 | 0.347 | 0.434 | 0.589 | 0.393 | 0.554 | 0.79 | 0.94 |
| Swin-Base | Acc. (avg.) | 86.2 | **86.3** | 99.7 | **100.0** | 99.4 | **99.5** | | |
| | MMD (neg.) | **-0.168** | -0.169 | -0.168 | **-0.086** | -0.169 | **-0.086** | 0.51 | 0.60 |
| | A-distance (neg.) | -2.00 | -2.00 | -2.00 | -2.00 | -2.00 | -2.00 | 0.0 | 0.0 |
| | **PAS (our)** | **0.361** | 0.349 | 0.399 | **0.589** | 0.374 | **0.56** | 0.62 | 0.89 |
| | Oracle* | 0.321 | 0.313 | 0.388 | 0.589 | 0.366 | 0.558 | 0.69 | 0.89 |

### (c) ImageCLEF

| Model | Method | C·I | C·P | I·C | I·P | P·C | P·I | Pearson | Spearman |
|---|---|---|---|---|---|---|---|---|---|
| ResNet-50 | Acc. (avg.) | **95.9** | 93.7 | **90.7** | 90.0 | 76.0 | **77.9** | | |
| | MMD (neg.) | -0.074 | -0.097 | -0.074 | **-0.022** | -0.097 | **-0.022** | -0.17 | -0.12 |
| | A-distance (neg.) | **-1.583** | -1.731 | -1.583 | **-0.807** | -1.731 | **-0.807** | -0.24 | -0.12 |
| | **PAS (our)** | **0.299** | 0.251 | 0.235 | **0.27** | 0.223 | **0.297** | 0.22 | **0.49** |
| | Oracle* | 0.287 | 0.243 | 0.195 | 0.254 | 0.111 | 0.2 | 0.84 | 0.71 |
| DeiT-Small | Acc. (avg.) | **97.5** | 97.5 | 93.7 | **95.2** | 78.3 | **80.8** | | |
| | MMD (neg.) | **-0.072** | -0.081 | -0.072 | **-0.02** | -0.081 | **-0.02** | -0.17 | -0.11 |
| | A-distance (neg.) | **-1.417** | -1.748 | -1.417 | **-0.807** | -1.748 | **-0.807** | -0.07 | -0.11 |
| | **PAS (our)** | **0.344** | 0.303 | 0.263 | **0.322** | 0.24 | **0.332** | 0.41 | 0.52 |
| | Oracle* | 0.333 | 0.293 | 0.239 | 0.31 | 0.169 | 0.25 | 0.83 | 0.83 |
| ViT-Base | Acc. (avg.) | **97.8** | 97.1 | **96.6** | 95.7 | 79.5 | **81.9** | | |
| | MMD (neg.) | -0.078 | -0.095 | -0.078 | **-0.022** | -0.095 | **-0.022** | -0.13 | -0.12 |
| | A-distance (neg.) | -1.483 | -1.714 | -1.483 | **-0.655** | -1.714 | **-0.655** | -0.14 | -0.12 |
| | **PAS (our)** | **0.399** | 0.359 | 0.304 | **0.377** | 0.262 | **0.363** | **0.55** | **0.54** |
| | Oracle* | 0.391 | 0.352 | 0.295 | 0.37 | 0.205 | 0.286 | 0.84 | 0.83 |

### (d) DomainNet

| Model | Method | C·P | C·R | C·S | P·C | P·R | P·S | R·C | R·P | R·S | S·C | S·P | S·R | Pearson | Spearman |
|---|---|---|---|---|---|---|---|---|---|---|---|---|---|---|---|
| ResNet-101 | Acc. (avg.) | 45.5 | 53.7 | **56.7** | 39.4 | **52.2** | 45.8 | 55.9 | **58.1** | 55.3 | **44.8** | 40.7 | 41.0 | 0.04 | 0.20 |
| | MMD (neg.) | -0.113 | -0.158 | **-0.079** | -0.113 | **-0.075** | -0.108 | -0.158 | -0.173 | **-0.079** | **-0.079** | -0.108 | -0.173 | 0.50 | 0.45 |
| | A-distance (neg.) | -1.789 | -1.73 | **-1.638** | -1.789 | **-1.656** | -1.777 | -1.73 | **-1.656** | -1.821 | **-1.638** | -1.777 | -1.821 | 0.58 | 0.53 |
| | **PAS (our)** | 0.108 | **0.145** | 0.088 | 0.08 | **0.159** | 0.083 | 0.128 | **0.184** | 0.107 | 0.088 | 0.098 | **0.114** | 0.58 | 0.53 |
| | Oracle* | -0.109 | -0.124 | -0.042 | -0.06 | -0.024 | -0.04 | 0.037 | 0.092 | 0.031 | -0.087 | -0.11 | -0.156 | 0.70 | 0.67 |
| DeiT-Small | Acc. (avg.) | 52.3 | **68.8** | 52.2 | 58.6 | **69.7** | 48.4 | **62.3** | 56.6 | 48.5 | 64.7 | 52.4 | **67.2** | 0.26 | 0.28 |
| | MMD (neg.) | -0.128 | -0.146 | **-0.088** | -0.128 | **-0.052** | -0.16 | -0.146 | **-0.052** | -0.186 | **-0.088** | -0.16 | -0.186 | 0.30 | 0.33 |
| | A-distance (neg.) | -1.784 | -1.734 | **-1.655** | -1.784 | **-1.639** | -1.768 | -1.734 | **-1.639** | -1.823 | **-1.655** | -1.768 | -1.823 | 0.30 | 0.33 |
| | **PAS (our)** | 0.13 | **0.152** | 0.093 | 0.091 | **0.175** | 0.086 | 0.139 | **0.218** | 0.11 | 0.096 | 0.117 | **0.127** | 0.39 | 0.57 |
| | Oracle* | -0.125 | -0.118 | -0.055 | -0.066 | -0.015 | -0.051 | 0.031 | 0.093 | 0.015 | -0.12 | -0.163 | -0.183 | -0.19 | -0.17 |
| DeiT-Base | Acc. (avg.) | 55.7 | **72.2** | 56.9 | 64.6 | **72.9** | 53.3 | **65.8** | 59.4 | 52.4 | 68.7 | 56.7 | **71.8** | 0.19 | 0.35 |
| | MMD (neg.) | -0.123 | -0.153 | **-0.095** | -0.123 | **-0.06** | -0.171 | -0.153 | **-0.06** | -0.227 | **-0.095** | -0.171 | -0.227 | **0.26** | 0.37 |
| | A-distance (neg.) | -1.796 | -1.746 | **-1.655** | -1.796 | **-1.682** | -1.79 | -1.746 | **-1.682** | -1.838 | **-1.655** | -1.79 | -1.838 | **0.26** | 0.44 |
| | **PAS (our)** | 0.126 | **0.147** | 0.086 | 0.085 | **0.165** | 0.079 | 0.137 | **0.211** | 0.102 | 0.089 | 0.119 | **0.112** | 0.26 | 0.44 |
| | Oracle* | -0.115 | -0.097 | -0.044 | -0.047 | 0.004 | -0.04 | 0.044 | 0.105 | 0.022 | -0.109 | -0.162 | -0.154 | -0.17 | -0.08 |
| ViT-Base | Acc. (avg.) | 60.7 | **77.7** | 60.7 | 66.2 | **75.9** | 57.2 | **69.7** | 64.6 | 57.9 | 71.4 | 62.7 | **76.3** | 0.07 | 0.15 |
| | MMD (neg.) | -0.14 | -0.157 | **-0.108** | -0.14 | **-0.116** | -0.175 | -0.157 | **-0.116** | -0.25 | **-0.108** | -0.175 | -0.25 | 0.18 | 0.21 |
| | A-distance (neg.) | -1.814 | -1.771 | **-1.686** | -1.814 | **-1.734** | -1.807 | -1.771 | **-1.734** | -1.859 | **-1.686** | -1.807 | -1.859 | 0.18 | 0.21 |
| | **PAS (our)** | 0.185 | **0.226** | 0.162 | 0.145 | **0.233** | 0.128 | 0.223 | **0.282** | 0.176 | 0.151 | **0.171** | 0.17 | **0.37** | **0.35** |
| | Oracle* | -0.003 | 0.018 | 0.042 | 0.012 | 0.066 | 0.007 | 0.135 | 0.184 | 0.103 | -0.021 | -0.075 | -0.062 | -0.13 | -0.08 |

Table 2: Statistics of the benchmarks used in the experiments.

| Dataset | #Samples | #Classes | Domains |
|---|---|---|---|
| Office-Home | 15,588 | 65 | **A** (Art), **C** (Clipart), **P** (Product), **R** (Real-world) |
| Office-31 | 4,110 | 31 | **A** (Amazon), **D** (DSLR), **W** (Webcam) |
| ImageCLEF | 1,800 | 12 | **C** (Caltech-256), **I** (ImageNet ILSVRC 2012), **P** (Pascal VOC 2012) |
| DomainNet | 569,010 | 345 | **C** (Clipart), **P** (Painting), **R** (Real), **S** (Sketch) |

Table 3: Correlation with the average target accuracy after adaptation. Showing Pearson correlation / Spearman's rank correlation.

| | Office-Home | Office-31 | ImageCLEF | DomainNet | Total |
|---|---|---|---|---|---|
| MMD | 0.55 / 0.51 | 0.45 / 0.53 | -0.14 / -0.08 | -0.09 / -0.03 | 0.37 / 0.37 |
| $\mathcal{A}$-distance | 0.32 / 0.17 | 0.26 / 0.35 | -0.13 / -0.07 | 0.07 / 0.06 | 0.04 / -0.16 |
| **PAS (our)** | **0.76 / 0.81** | **0.63 / 0.78** | **0.44 / 0.60** | **0.53 / 0.56** | **0.83 / 0.88** |
| Oracle* | 0.89 / 0.90 | 0.71 / 0.86 | 0.78 / 0.85 | 0.21 / 0.21 | 0.88 / 0.91 |

centroid that is not the true class. In the ideal case where the closest class centroid is the actual class of the sample, the oracle is the same as **PAS**, otherwise, the oracle value is smaller. The oracle validates the existing relationship between the clusters distance and the target accuracy.

## 4.1 SELECTION OF THE SOURCE DOMAIN

The results for the four benchmark datasets are presented in Table 1 (a) - (d). For each source-target pair in the benchmarks, we group the domain adaptation methods using the same pre-trained feature extractor and report their average target accuracy, followed by the baselines and our **PAS** score. We highlight the highest values among the different choices of source domains. We also report the correlation (Pearson and Spearman's rank correlation) between the average target accuracy and the scores. The detailed results for each individual domain adaptation method are presented in the Supplementary Material A.1.

We report in Table 3 the overall correlation for all scenarios of each benchmark (all target domains, source domains and pre-trained models). The results show that the **PAS** score is strongly correlated with target accuracy. We observe an overall Spearman's rank correlation of **0.88** over all the results.

The most important results are reported in Table 4, where we present the correlation for each target domain. This correlation is the most useful for users in real-world scenarios. Given a target domain of interest and many options of source domains and pre-trained models, we show that our **PAS** score has a strong correlation with the final target accuracy. The empirical results indicate that our proposed **PAS** score is effective in selecting the best source domain among many candidates.

We summarize our results in Figure 3. Each box in the graph represents the target accuracy of different domain adaptation methods using the same pre-trained backbone for a source-target domains pair. We observe that higher **PAS** values are consistently related to high accuracy on the target domain. This indicates that **PAS** may be useful not only for selecting the most appropriate source domain, but also to estimate beforehand the success of the domain adaptation.

Table 4: Correlation with the average target accuracy after adaptation for each target domain. Each cell considers the results for a target domain and all available source domains and pre-trained models. Showing Pearson correlation / Spearman's rank correlation.

| | Office-Home | | | | Office-31 | | | ImageCLEF | | | DomainNet | | | |
|---|---|---|---|---|---|---|---|---|---|---|---|---|---|---|
| | A | C | P | R | A | D | W | C | I | P | C | P | R | S |
| MMD | 0.41 / 0.26 | 0.28 / 0.21 | 0.41 / 0.29 | 0.25 / 0.21 | -0.02 / -0.15 | 0.44 / 0.60 | 0.45 / 0.36 | 0.54 / 0.49 | -0.03 / -0.22 | 0.61 / 0.60 | -0.56 / -0.42 | 0.33 / 0.20 | 0.23 / 0.47 | -0.38 / -0.42 |
| $\mathcal{A}$-distance | 0.12 / -0.13 | -0.46 / -0.43 | 0.15 / -0.09 | -0.05 / -0.26 | -0.19 / -0.31 | 0.27 / 0.28 | 0.14 / 0.07 | 0.43 / 0.38 | 0.10 / 0.05 | 0.64 / 0.60 | 0.09 / -0.04 | 0.35 / 0.17 | 0.27 / 0.30 | -0.10 / -0.32 |
| **PAS (our)** | **0.70 / 0.70** | **0.81 / 0.78** | **0.79 / 0.74** | **0.75 / 0.75** | **0.65 / 0.81** | **0.70 / 0.81** | **0.70 / 0.75** | **0.82 / 0.76** | **0.73 / 0.66** | **0.87 / 0.83** | **0.71 0.67** | **0.75 / 0.76** | **0.59 / 0.71** | **0.48 / 0.35** |
| Oracle* | 0.82 / 0.85 | 0.91 / 0.90 | 0.84 / 0.81 | 0.80 / 0.87 | 0.74 / 0.88 | 0.73 / 0.90 | 0.74 / 0.78 | 0.81 / 0.76 | 0.74 / 0.66 | 0.97 / 0.94 | 0.36 / 0.50 | 0.71 / 0.62 | 0.61 / 0.72 | 0.28 / 0.33 |

The results on the *ImageCLEF* benchmark illustrate the scenarios where the **PAS** score is not effective. This benchmark (especially the *P* domain) contains images with multiple objects. In many cases, the sample is very close to the centroid of one class that is indeed present in the image, but the true label is related to another object in the scene. In these cases, the **PAS** for the sample is high, showing a high similarity with one source class, but the final accuracy is low, as the sample is classified as the wrong class. We show examples in the supplementary material A.2

## 4.2 THE SELECTION OF THE PRE-TRAINED FEATURE EXTRACTOR

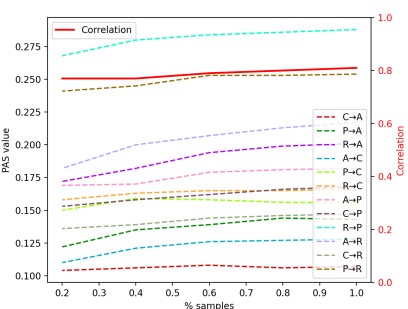

Figure 5: The **PAS** value varying with the number of samples for the *Office-Home*. The **PAS** values are quite robust to varying numbers of samples. Most importantly, the relative order of PAS values for different source domains remains unchanged.

The results in the literature presented in table 1 compare methods with different backbones and demonstrate that **PAS** can be applied to select the most suitable pre-trained feature extractor. However, they do not consider the impact of different pre-trained feature extractors over the same domain adaptation method. For analyzing the robustness of **PAS** over different choices of pre-trained methods, we keep the domain adaptation method fixed and vary the pre-trained backbone. We select two of the most challenging domain adaptation scenarios: the *A→C* adaptation in the *Office-Home* benchmark and *W→A* adaptation in the *Office-31* benchmark. We show results for two popular domain adaptation methods: *DANN* Ganin et al. (2016) and *MCC* Jin et al. (2020). We train each method following the code provided by the *Tllib* library Jiang et al. (2022); Junguang Jiang (2020) with the default hyperparameters. The results are shown in figure 4. Higher **PAS** values are attributed to pre-trained models that lead to higher target accuracy before performing domain adaptation, indicating that our score

may be applied for the selection of the pre-trained model.

## 4.3 THE IMPACT OF THE SAMPLE SIZE

The time complexity of the PAS computation is linear in the number of samples. This can be limiting for a quick evaluation of larger datasets and scenarios with many candidate source domains.

To optimize the computation time, we show that our score can be calculated using only a subset of the samples. We randomly select a subset of the samples of both source and target domains. The results are presented in the figure 5. The **PAS** values are quite robust to varying numbers of samples. Most importantly, the relative order of PAS values for different source domains remains unchanged.

## 4.4 DESIGN CHOICES

The **PAS** score considers the cosine distance between each target sample and the source class centroids. We experimentally evaluated alternative design choices and compare the correlation between the score and the target accuracy. The results are shown in table 5 and demonstrate how the overall correlation between the target accuracy and **PAS**, as proposed, is higher.

We change **PAS** to consider the Euclidean distance instead of the cosine distance. In the domain adaptation setting, the cosine distance has advantages over using the Euclidean distance in the original latent space, as it ignores the magnitude of

|  | Office-Home | Office-31 | ImageCLEF | DomainNet | Total |  |
| --- | --- | --- | --- | --- | --- | --- |
| **PAS** | **0.76** | 0.63 | **0.44** |  | **0.58** | **0.79** |
| Euclidean distance | 0.70 | **0.69** | 0.27 | 0.54 | 0.68 |  |
| Average cosine distance | 0.66 | 0.52 | 0.12 | 0.48 | 0.66 |  |

Table 5: Pearson correlation between the target accuracy and the **PAS** score, which considers the cosine distance to the cluster centroid, and modifications using the Euclidean distance to the centroid and the average cosine distance to the source cluster samples. The maximum correlation value for each benchmark is highlighted. The design choices of **PAS** lead to the higher overall correlation between the score and the target accuracy.

representations (e.g., a difference in the illumination in images that reflects on the intensity of the features detected by the model) and focuses only on the differences in the angles (the difference between classes). Also, the cosine distance is less affected by the high-dimensionality of the data (the phenomenon known as *curse of dimensionality* Bellman (1966)).

We also modify **PAS** to use the average pairwise distance to the source samples instead of the distance to the source cluster centroid. The pairwise distance is a good summarization of the closeness of the target sample to the source samples of the class. On the other hand, the distance to the centroid measures how well the target sample is aligned to the dimensions of greatest alignment between the samples in the cluster, as the centroid formulation $1/|\mathcal{D}^S| \sum_{i=1}^{|\mathcal{D}^S|} x_i^S$ makes samples pointing in similar directions add up in that direction.

## 5 CONCLUSION AND FUTURE WORK

We present **Potential Adaptability Score (PAS)**, a new score to select, among many candidates, the source domain or pre-trained model that are likely to lead to the best target accuracy when used for unsupervised domain adaptation. We evaluate our score on four of the most popular benchmarks for domain adaptation and show that it has a high correlation with the target accuracy and selects the best source domain in most cases. We also show that **PAS** can be computed more efficiently with fewer samples.

We suggest two improvements for future work. Although our score could be applied to any classification task, we focus on vision problems, specifically the image classification task, which is the most common task in the domain adaptation literature. Showing its efficacy on other modalities and tasks demands the availability of a diverse set of benchmarks and specialized domain adaptation methods. Also, we focus on the single-source domain adaptation problem, where only a single source domain is considered during the training. Future works may extend our work to select multiple source domains, in the setting known as multi-source domain adaptation.

## ETHICS STATEMENT

Although our work does not directly address issues of social harm, we acknowledge that our **PAS** score is not immune to bias and fairness concerns. If a biased model achieves higher accuracy on the target data, our framework is likely to select it as the best pre-trained model for domain adaptation. In our experiments, we mitigate such risks by focusing on pre-trained models and benchmarks that are widely adopted within the research community.

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

## A  Supplementary Material

### A.1  Results

The tables 6, 7, 8, and 9 show the extended results of table 1. The accuracy for each method is listed, as well as its accuracy correlation with the scores.

### A.2  Examples of samples in the *ImageCLEF* benchmark

We present some examples of when the **PAS** score fails to predict the target accuracy. The figure 6 shows examples of misclassified images from the **P** (*Pascal VOC 2012*) domain of the *ImageCLEF* benchmark. Many images contain more than one object. The sample may be very similar to a class present in the image. However, the true class refers to another object also contained in the image. In such cases, the **PAS** value is high, but the accuracy is low.

Table 6: Target accuracy of domain adaptation methods and transferability scores for the ***Office-Home*** dataset. The highest values are highlighted. * Oracle baseline that considers the target labels.

| | Target Source | A C | A P | A R | C A | C P | C R | P A | P C | P R | R A | R C | R P | Pearson | Spearman |
|---|---|---|---|---|---|---|---|---|---|---|---|---|---|---|---|
| ResNet-50 | DAN | 57.7 | 54.9 | 66.2 | 45.6 | 40.0 | 49.1 | 67.7 | 63.8 | 77.9 | 73.9 | 66.0 | 74.5 | 0.74 | 0.81 |
| | DANN | 55.8 | 55.8 | 71.1 | 53.8 | 55.1 | 60.7 | 62.6 | 67.3 | 81.1 | 74.0 | 67.3 | 77.9 | 0.91 | 0.85 |
| | ADDA | 59.7 | 61.4 | 71.1 | 52.6 | 52.5 | 58.6 | 62.9 | 68.0 | 80.2 | 74.0 | 68.8 | 77.6 | 0.84 | 0.80 |
| | JAN | 60.6 | 60.5 | 71.0 | 50.8 | 49.6 | 55.9 | 71.9 | 68.3 | 80.5 | 76.5 | 68.7 | 76.9 | 0.78 | 0.81 |
| | CDAN | 62.0 | 62.4 | 75.5 | 55.2 | 54.3 | 61.0 | 72.4 | 69.7 | 83.8 | 77.6 | 70.9 | 80.5 | 0.85 | 0.83 |
| | MCD | 63.7 | 61.5 | 74.5 | 51.7 | 52.8 | 58.4 | 72.2 | 69.5 | 81.8 | 78.2 | 70.8 | 78.0 | 0.78 | 0.83 |
| | BSP | 61.0 | 60.9 | 73.4 | 54.7 | 55.2 | 60.3 | 67.7 | 69.4 | 81.2 | 76.2 | 70.9 | 80.2 | 0.85 | 0.80 |
| | AFN | 65.0 | 65.0 | 72.3 | 53.2 | 51.4 | 57.8 | 72.7 | 71.3 | 82.4 | 76.8 | 72.3 | 77.9 | 0.73 | 0.78 |
| | MDD | 63.5 | 62.5 | 73.5 | 56.2 | 54.8 | 60.9 | 75.4 | 72.1 | 84.5 | 79.6 | 73.8 | 79.9 | 0.80 | 0.78 |
| | MCC | 67.5 | 66.6 | 74.4 | 58.4 | 54.8 | 61.4 | 79.6 | 77.0 | 85.6 | 83.0 | 78.5 | 81.8 | 0.70 | 0.76 |
| | FixMatch | 65.3 | 67.2 | 74.9 | 56.4 | 56.4 | 63.5 | 76.4 | 73.8 | 84.3 | 79.9 | 71.2 | 80.6 | 0.81 | 0.87 |
| | Avg. | 62.0 | 61.7 | **72.5** | 53.5 | 52.4 | **58.9** | 71.0 | 70.0 | **82.1** | 77.2 | 70.8 | **78.7** | **0.81** | **0.82** |
| | **PAS (our)** | 0.107 | 0.143 | **0.201** | 0.128 | 0.156 | **0.166** | 0.182 | 0.168 | **0.288** | 0.217 | 0.147 | **0.254** | | |
| DeiT-Small | TRANS-DA | 69.7 | 68.6 | 73.5 | 57.7 | 56.3 | 58.5 | 80.8 | 83 | 85 | 81.5 | 80.1 | 81.5 | 0.69 | 0.79 |
| | WinTR | 76.8 | 73.4 | 77.2 | 65.3 | 60 | 63.1 | 84.1 | 84.5 | 86.8 | 85 | 84.4 | 85.7 | 0.64 | 0.78 |
| | DOT | 74.9 | 72.4 | 76.4 | 63.7 | 61 | 64.1 | 82.2 | 84.3 | 86.7 | 84.3 | 83 | 84.8 | 0.68 | 0.79 |
| | CDTrans | 75.6 | 72.5 | 77 | 60.6 | 56.7 | 59.1 | 79.5 | 81 | 85.5 | 82.4 | 82.3 | 84.4 | 0.63 | 0.76 |
| | Avg. | 74.3 | 71.7 | **76.0** | **61.8** | 58.5 | 61.2 | 81.7 | 83.2 | **86** | 83.3 | 82.5 | **84.1** | **0.67** | **0.78** |
| | **PAS (our)** | 0.143 | 0.183 | **0.25** | 0.175 | 0.186 | **0.204** | 0.261 | 0.221 | **0.348** | 0.295 | 0.2 | **0.301** | | |
| DeiT-Base | DOT | 80 | 78.2 | 79.7 | 69 | 65.4 | 67.3 | 85.6 | 85.2 | 89.3 | 87 | 86.4 | 87.9 | 0.66 | 0.75 |
| | CDTrans | 81.5 | 79.6 | 82 | 68.8 | 63.3 | 66 | 85 | 87.1 | 90.6 | 86.9 | 87.3 | 88.2 | 0.62 | 0.73 |
| | PMTrans | 83 | 78.5 | 81.7 | 71.8 | 67.4 | 70.7 | 87.3 | 87.7 | 92 | 88.3 | 87.8 | 89.3 | 0.67 | 0.73 |
| | Avg. | **81.5** | 78.8 | 81.1 | **69.9** | 65.4 | 68.0 | 86.0 | 86.7 | **90.6** | 87.4 | 87.2 | **88.5** | **0.65** | **0.73** |
| | **PAS (our)** | 0.138 | 0.176 | **0.243** | 0.166 | 0.172 | **0.194** | 0.245 | 0.209 | **0.339** | 0.287 | 0.193 | **0.295** | | |
| ViT-Small | SSRT | 79.9 | 80.7 | 82 | 67 | 66 | 69.4 | 84.2 | 84.3 | 89.9 | 88.3 | 87.6 | 88.3 | 0.69 | 0.84 |
| | SAMB | 80.2 | 78.8 | 82.4 | 65.7 | 64.4 | 67 | 84 | 84.1 | 88 | 87.7 | 86.7 | 88.6 | 0.67 | 0.82 |
| | Avg. | 80.1 | 79.8 | **82.2** | 66.4 | 65.2 | **68.2** | 84.1 | 84.2 | **89.0** | 88.0 | 87.2 | **88.5** | **0.68** | **0.83** |
| | **PAS (our)** | 0.172 | 0.198 | **0.262** | 0.182 | 0.199 | **0.217** | 0.251 | 0.235 | **0.357** | 0.294 | 0.219 | **0.316** | | |
| ViT-Base | SAMB | 80.8 | 81.6 | 84.1 | 68.7 | 68.7 | 70.9 | 85 | 86 | 91.1 | 88.9 | 88.3 | 90.2 | 0.77 | 0.88 |
| | DoT | 81.8 | 81.2 | 82.9 | 72.9 | 70.6 | 72.2 | 89.8 | 89.6 | 90.8 | 90.3 | 90.1 | 92.4 | 0.75 | 0.84 |
| | TVT | 77.4 | 75.6 | 79.1 | 67.1 | 64.9 | 67.2 | 83.5 | 85 | 88 | 87.3 | 85.6 | 86.6 | 0.78 | 0.85 |
| | SSRT | 85.1 | 85 | 85.7 | 75.2 | 74.2 | 78.6 | 89 | 88.3 | 91.8 | 91.1 | 90 | 91.3 | 0.76 | 0.87 |
| | BCAT | 84.2 | 84.1 | 85.7 | 74.2 | 74.5 | 74.8 | 90.6 | 90.9 | 92.2 | 90.9 | 89.9 | 90.8 | 0.74 | 0.83 |
| | PMTrans | 88.9 | 88.5 | 89.5 | 81.2 | 80 | 82.4 | 91.6 | 91.6 | 94.5 | 92.4 | 93 | 93.4 | 0.74 | 0.84 |
| | Avg. | 83.0 | 82.7 | **84.5** | 73.2 | 72.2 | **74.4** | 88.3 | 88.6 | **91.4** | 90.2 | 89.5 | **90.8** | **0.76** | **0.85** |
| | **PAS (our)** | 0.254 | 0.28 | **0.357** | 0.262 | 0.271 | **0.296** | 0.361 | 0.339 | **0.462** | 0.405 | 0.316 | **0.417** | | |
| Swin-Base | PMTrans | 88.4 | 87.9 | 89 | 81.3 | 80.4 | 80.9 | 92.9 | 93.4 | 93.4 | 92.8 | 93.2 | 93 | 0.75 | 0.72 |
| | BCAT | 88.6 | 87.4 | 86.7 | 75.3 | 73.7 | 75.4 | 90 | 90.3 | 93.5 | 92.9 | 92.7 | 92.5 | 0.68 | 0.74 |
| | Avg. | **88.5** | 87.7 | 87.9 | **78.3** | 77.1 | 78.2 | 91.5 | 91.9 | **94.2** | 92.9 | **93.0** | 92.8 | **0.72** | **0.72** |
| | **PAS (our)** | 0.232 | 0.251 | **0.327** | 0.231 | 0.244 | **0.269** | 0.323 | 0.318 | **0.43** | 0.37 | 0.294 | **0.384** | | |

Table 7: Target accuracy of domain adaptation methods and transferability scores for the ***Office-31*** dataset. The highest values are highlighted. * Oracle baseline that considers the target labels.

| | Target | A | | D | | W | | Correlation with **PAS** | |
| | Source | D | W | A | W | A | D | Pearson | Spearman |
|---|---|---|---|---|---|---|---|---|---|
| ResNet-50 | DANN | 73.3 | 70.4 | 83.6 | 100.0 | 91.4 | 97.9 | 0.78 | 0.66 |
| | ADDA | 69.6 | 72.5 | 90.0 | 99.7 | 94.6 | 97.5 | 0.67 | 0.60 |
| | BSP | 74.1 | 73.8 | 88.2 | 100.0 | 92.7 | 97.9 | 0.75 | 0.66 |
| | DAN | 66.9 | 65.2 | 87.3 | 100.0 | 84.2 | 98.4 | 0.83 | 0.83 |
| | JAN | 69.2 | 71.0 | 89.4 | 100.0 | 93.7 | 98.4 | 0.70 | 0.60 |
| | CDAN | 73.4 | 70.4 | 89.9 | 100.0 | 93.8 | 98.5 | 0.71 | 0.66 |
| | MCD | 68.3 | 67.6 | 87.3 | 100.0 | 90.4 | 98.5 | 0.76 | 0.66 |
| | AFN | 72.9 | 71.1 | 94.4 | 100.0 | 94.0 | 98.9 | 0.67 | 0.83 |
| | MDD | 76.6 | 72.2 | 94.4 | 100.0 | 95.6 | 98.6 | 0.65 | 0.66 |
| | MCC | 75.5 | 74.2 | 95.6 | 99.8 | 94.1 | 98.4 | 0.66 | 0.83 |
| | FixMatch | 70.0 | 68.1 | 95.4 | 100.0 | 86.4 | 98.2 | 0.75 | 0.83 |
| | Avg. | **71.1** | 70.0 | 89.6 | **99.9** | 90.6 | **98.1** | **0.73** | **0.66** |
| | **PAS (our)** | **0.265** | 0.239 | 0.286 | **0.454** | 0.236 | **0.423** | | |
| DeiT-Small | TRANS-DA | 77 | 77.1 | 94.8 | 100 | 95.8 | 98.8 | 0.69 | 0.71 |
| | CDTrans | 78.4 | 78 | 94.6 | 99.6 | 93.5 | 98.2 | 0.74 | 0.94 |
| | Avg. | **77.7** | 77.6 | 94.7 | **99.8** | 94.65 | **98.5** | **0.72** | **0.94** |
| | **PAS (our)** | **0.283** | 0.266 | 0.304 | **0.472** | 0.278 | **0.447** | | |
| DeiT-Base | CDTrans | 81.1 | 81.9 | 97 | 100 | 96.7 | 99 | 0.69 | 0.83 |
| | PMTrans | 81.4 | 82.1 | 96.5 | 100 | 99 | 99.4 | 0.64 | 0.71 |
| | Avg. | 81.3 | **82.0** | 96.8 | **100.0** | 97.9 | **99.2** | **0.66** | **0.71** |
| | **PAS (our)** | **0.268** | 0.241 | 0.304 | **0.443** | 0.251 | **0.418** | | |
| ViT-Small | SSRT | **83.5** | 82.2 | 98.6 | **100** | 97.7 | **99.2** | **0.61** | **0.94** |
| | **PAS (our)** | **0.283** | 0.27 | 0.302 | **0.509** | 0.276 | **0.473** | | |
| ViT-Base | DoT | 85.1 | 86.8 | 96.7 | 100 | 96.6 | 99.4 | 0.74 | 0.83 |
| | TVT | 84.9 | 86.1 | 96.4 | 100 | 96.4 | 99.4 | 0.75 | 0.75 |
| | SSRT | 79.2 | 79.9 | 95.8 | 100 | 95.7 | 99.2 | 0.72 | 0.83 |
| | BCAT | 84.9 | 85.8 | 97.5 | 100 | 96.1 | 99.1 | 0.73 | 0.83 |
| | PMTrans | 85.7 | 86.3 | 99.4 | 100 | 99.1 | 99.6 | 0.62 | 0.83 |
| | Avg. | 84.0 | **85.0** | 97.2 | **100.0** | 96.8 | **99.3** | **0.71** | **0.83** |
| | **PAS (our)** | **0.423** | 0.395 | 0.453 | **0.59** | 0.412 | **0.558** | | |
| Swin-Base | PMTrans | 86.7 | 86.5 | 99.8 | 100 | 99.5 | 99.4 | 0.61 | 0.83 |
| | BCAT | 85.7 | 86.1 | 99.6 | 100 | 99.2 | 99.5 | 0.63 | 0.89 |
| | Avg. | 86.2 | **86.3** | 99.7 | **100** | 99.4 | **99.5** | **0.62** | **0.89** |
| | **PAS (our)** | 0.361 | 0.349 | 0.399 | 0.589 | 0.374 | 0.56 | | |

Table 8: Target accuracy of domain adaptation methods and transferability scores for the *Image-CLEF* dataset. The highest values are highlighted. * Oracle baseline that considers the target labels.

| | Target | C | | I | | P | | Correlation with PAS | |
| | Source | I | P | C | P | C | I | Pearson | Spearman |
|---|---|---|---|---|---|---|---|---|---|
| | RTN | 95.3 | 92.2 | 86.9 | 86.8 | 72.7 | 75.6 | 0.29 | 0.49 |
| | MADA | 96.0 | 92.2 | 88.8 | 87.9 | 75.2 | 75.0 | 0.20 | 0.26 |
| | iCAN | 94.7 | 92 | 89.9 | 89.7 | 78.5 | 79.5 | 0.23 | 0.49 |
| | CDAN-E | 97.7 | 94.3 | 91.3 | 90.7 | 74.2 | 77.7 | 0.27 | 0.49 |
| | SymNets | 97.0 | 96.4 | 93.4 | 93.6 | 78.7 | 80.2 | 0.17 | 0.60 |
| | MEDA | 95.7 | 95.5 | 92.2 | 92.5 | 78.5 | 79.7 | 0.16 | 0.60 |
| ResNet-50 | SPL | 96.7 | 96.3 | 95.7 | 94.5 | 80.5 | 78.3 | 0.02 | 0.26 |
| | DS-c | 92.8 | 91.3 | 87.3 | 86.7 | 70.4 | 78.7 | 0.39 | 0.49 |
| | CAN | 95.5 | 95.2 | 91.6 | 91.8 | 76.4 | 78.5 | 0.19 | 0.60 |
| | JAN | 94.7 | 91.7 | 89.5 | 88.0 | 74.2 | 76.8 | 0.24 | 0.49 |
| | CDAN | 98.3 | 94 | 90.7 | 88.3 | 76.7 | 77.2 | 0.22 | 0.49 |
| | Avg. | **95.9** | 93.7 | **90.7** | 90.0 | 76.0 | **77.9** | **0.22** | **0.49** |
| | **PAS (our)** | **0.299** | 0.251 | 0.235 | **0.27** | 0.223 | **0.297** | | |
| DeiT-small | TRANS-DA | **97.5** | **97.5** | 93.7 | **95.2** | 78.3 | **80.8** | 0.41 | 0.52 |
| | **PAS (our)** | **0.344** | 0.303 | 0.263 | **0.322** | 0.24 | **0.332** | | |
| | VT-ADA | 97.3 | 96.0 | 96.2 | 94.1 | 78.9 | 81.8 | 0.55 | 0.49 |
| | CSTrans | 98.2 | 98.2 | 97.0 | 97.2 | 80.0 | 82.0 | 0.54 | 0.62 |
| ViT-Base | Avg. | **97.8** | 97.1 | **96.6** | 95.7 | 79.5 | **81.9** | **0.55** | **0.54** |
| | **PAS (our)** | **0.399** | 0.359 | 0.304 | **0.377** | 0.262 | **0.363** | | |

Table 9: Target accuracy of domain adaptation methods and transferability scores for the *DomainNet* dataset. The highest values are highlighted. * Oracle baseline that considers the target labels.

| | Target | C | | | P | | | R | | | S | | | Correlation with PAS | |
| | Source | P | R | S | C | R | S | C | P | S | C | P | R | Pearson | Spearman |
|---|---|---|---|---|---|---|---|---|---|---|---|---|---|---|---|
| | DAN | 45.9 | 50.8 | 56.1 | 38.8 | 49.8 | 45.9 | 55.2 | 59.0 | 55.5 | 43.9 | 40.8 | 38.9 | 0.54 | 0.49 |
| | DANN | 41.7 | 50.7 | 55.0 | 37.9 | 50.8 | 45.0 | 54.3 | 55.6 | 54.5 | 44.4 | 36.8 | 40.1 | 0.53 | 0.46 |
| | JAN | 47.2 | 54.2 | 56.6 | 40.5 | 52.6 | 46.2 | 56.7 | 59.9 | 55.5 | 45.1 | 43.0 | 41.9 | 0.63 | 0.58 |
| | CDAN | 45.1 | 55.6 | 57.2 | 40.4 | 53.6 | 46.4 | 56.8 | 58.4 | 55.7 | 46.1 | 40.5 | 43.0 | 0.60 | 0.50 |
| ResNet-50 | MCD | 44.6 | 52.0 | 55.5 | 37.5 | 51.5 | 44.6 | 52.9 | 54.5 | 52.0 | 44.0 | 41.6 | 39.7 | 0.57 | 0.47 |
| | MDD | 48.6 | 58.3 | 58.7 | 42.9 | 53.7 | 46.5 | 59.5 | 59.4 | 57.7 | 47.5 | 42.6 | 46.2 | 0.60 | 0.59 |
| | MCC | 45.4 | 54.4 | 58.1 | 37.7 | 53.1 | 46.3 | 55.7 | 59.8 | 56.2 | 42.6 | 39.9 | 37.0 | 0.57 | 0.43 |
| | Avg. | 45.5 | 53.7 | **56.7** | 39.4 | **52.2** | 45.8 | 55.9 | **58.1** | 55.3 | **44.8** | 40.7 | 41.0 | **0.58** | **0.53** |
| | **PAS (our)** | 0.108 | **0.145** | 0.088 | 0.08 | **0.159** | 0.083 | 0.128 | **0.184** | 0.107 | 0.088 | 0.098 | **0.114** | | |
| | WinTR | 53.2 | 70.5 | 51.6 | 62.0 | 71.3 | 50.1 | 63.1 | 55.9 | 48.8 | 65.3 | 54.1 | 70.1 | 0.32 | 0.54 |
| | DOT | 51.3 | 67.6 | 51.7 | 58.5 | 70.4 | 47.2 | 62.3 | 57 | 49.4 | 64.6 | 49.9 | 65.4 | 0.42 | 0.52 |
| DeiT-Small | CDTRANS | 52.5 | 68.3 | 53.2 | 55.4 | 67.4 | 48 | 61.5 | 56.8 | 47.2 | 64.3 | 53.2 | 66.2 | 0.41 | 0.55 |
| | Avg. | 52.3 | **68.8** | 52.2 | 58.6 | **69.7** | 48.4 | **62.3** | 56.6 | 48.5 | 64.7 | 52.4 | **67.2** | **0.39** | **0.57** |
| | **PAS (our)** | 0.13 | **0.152** | 0.093 | 0.091 | **0.175** | 0.086 | 0.139 | **0.218** | 0.11 | 0.096 | 0.117 | **0.127** | | |
| | DOT | 53.6 | 71.2 | 55.2 | 61.8 | 72.2 | 50.5 | 62.9 | 56.9 | 49.3 | 67.3 | 52.9 | 69.8 | 0.27 | 0.45 |
| | CDTRANS | 57.2 | 72.6 | 58.1 | 62.9 | 72.1 | 53.9 | 66.2 | 61.5 | 52.9 | 69.0 | 59.0 | 72.5 | 0.33 | 0.43 |
| DeiT-Base | WINTR | 56.3 | 72.8 | 57.3 | 69.2 | 74.4 | 55.6 | 68.2 | 59.8 | 55.1 | 69.9 | 58.1 | 73.1 | 0.20 | 0.39 |
| | Avg | 55.7 | **72.2** | 56.9 | 64.6 | **72.9** | 53.3 | **65.8** | 59.4 | 52.4 | 68.7 | 56.7 | **71.8** | **0.26** | **0.44** |
| | **PAS (our)** | 0.126 | **0.147** | 0.086 | 0.085 | **0.165** | 0.079 | 0.137 | **0.211** | 0.102 | 0.089 | **0.119** | 0.112 | | |
| | SAMB | 60.5 | 77.8 | 61.8 | 63.8 | 77.1 | 56.8 | 68 | 64.7 | 58.4 | 71.1 | 64 | 77.5 | 0.38 | 0.43 |
| | DoT | 61.3 | 79.6 | 60.4 | 73.2 | 79.2 | 59.7 | 71.1 | 63.2 | 56.4 | 72.6 | 61.9 | 78.3 | 0.24 | 0.31 |
| ViT-Base | SSRT | 60.2 | 75.8 | 59.8 | 61.7 | 71.4 | 55.2 | 69.9 | 66.0 | 58.9 | 70.6 | 62.2 | 73.2 | 0.50 | 0.46 |
| | Avg. | 60.7 | **77.7** | 60.7 | 66.2 | **75.9** | 57.2 | **69.7** | 64.6 | 57.9 | 71.4 | 62.7 | **76.3** | **0.37** | **0.35** |
| | **PAS (our)** | 0.185 | **0.226** | 0.162 | 0.145 | **0.233** | 0.128 | 0.223 | **0.282** | 0.176 | 0.151 | **0.171** | 0.17 | | |

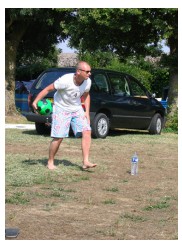

(a) Predicted: Car |True: Bottle

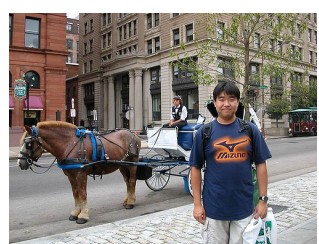

(b) Predicted: Horse |True: Person

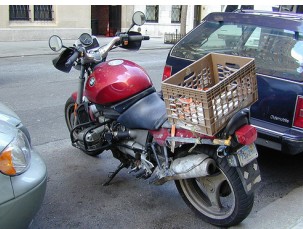

(c) Predicted: Motorcycle |True: Car

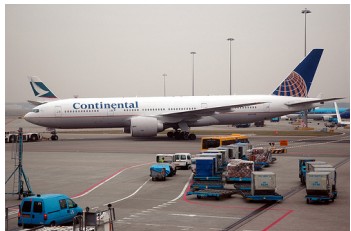

(d) Predicted: Plane |True: Bus

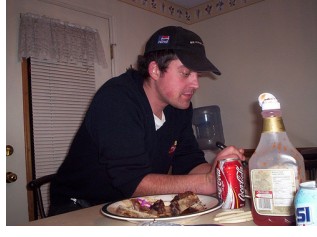

(e) Predicted: Person |True: Bottle

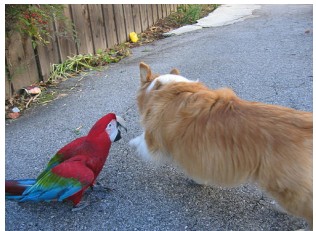

(f) Predicted: Dog |True: Bird

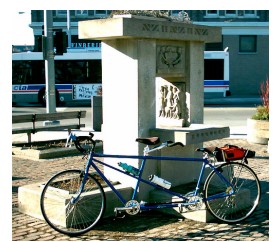

(g) Predicted: Bike |True: Bus

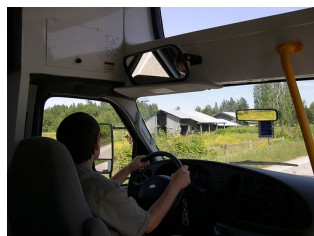

(h) Predicted: Car |True: Person

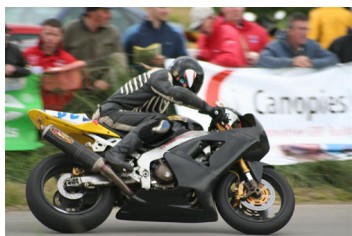

(i) Predicted: Motorcycle |True: Person

Figure 6: Examples of images misclassified by the domain adaptation method *DANN* in the dataset *P* (*Pascal VOC 2012*) of the *ImageCLEF* benchmark

