# OpenReview forum: "PAS: Estimating the target accuracy before domain adaptation"
_ICLR.cc/2026/Conference — ICLR 2026 Poster_

### Official Review · Reviewer_mW8M · 2025-10-20

**Soundness:** 3
**Presentation:** 2
**Contribution:** 2
**Rating:** 4
**Confidence:** 4

**Summary:**

This paper studies an interesting and important problem in transfer learning, specifically unsupervised domain adaptation (UDA) — the assessment of source model transferability. To address this, the authors propose a Potential Adaptability Score (PAS), which measures the compatibility between source and target domains based on the generalization capacity of pre-trained models. PAS leverages pre-trained feature embeddings to quantify the potential adaptability across domains. Although the proposed metric is conceptually interesting and somewhat novel, a major concern lies in the limited discussion and comparison with existing transferability indicators. The authors claim that no comparable baselines exist. However, several existing indices, though possibly less effective, could serve as meaningful references, for example feature distribution discrepancy measures (Maximum Mean Discrepancy, Wasserstein distance) and prediction uncertainty.

**Strengths:**

-	The paper is well organized and clearly written.
-	The paper targets transferability in domain adaptation, which is both interesting and relevant to the ICLR community.
-	The proposed PAS index provides a potentially useful tool for estimating transferability in UDA scenarios.

**Weaknesses:**

-	Some claims in the manuscript appear overstated. For instance, Lines 108–109 state that “this is the first proposal for transferability estimation for domain adaptation.” However, there exist several prior works on transferability estimation, such as NCE, LEEP, and LogME. While these studies are mentioned in the related work section, the claim of being the first proposal is inaccurate, as these methods also target transferability and can be adapted for domain adaptation settings.

-	Another overstatement appears in the experimental design section (Line 313), where the authors argue that “there is no comparable baseline.” This assumption is not entirely fair. Several existing metrics, such as Maximum Mean Discrepancy, Wasserstein distance, and the averaged prediction entropy over target data, could serve as comparable baselines to evaluate the proposed PAS. Including these in the experiments would make the comparison more convincing and substantiate the claimed superiority of PAS.

-	The paper would benefit from a deeper analysis, particularly in cases where PAS fails to identify the most suitable pre-trained model. Discussing such failure cases could provide valuable insights into the limitations and applicability of the proposed metric.

**Questions:**

Please refer to the weakness section.

---

> ### Author Response · Authors · 2025-11-18
>
> Thank you for your thoughtful feedback on our work. We hope to address the concerns that have been raised.
>
> __W1: Prior works on transferability estimation__
>
> To the best of our knowledge, prior works on transferability estimation, such as NCE, LEEP, and LogME, focus on the transfer learning problem, where one is given a model pre-trained on a source task and a labeled target dataset from a (probably different) downstream task. In the unsupervised domain adaptation setting, two different domains (source and target) are given for the downstream task, and the target samples are unlabeled. We do believe that such existing methods could give valuable inspiration to future works on transferability estimation on the domain adaptation scenario. However, we are not sure how they could be directly adapted for domain adaptation as originally proposed. Do you have any suggestions on how we could adapt such methods?
>
> __W2: Comparison to other baselines__
>
> As we mentioned to reviewer YdsH, distance metrics like Maximum Mean Discrepancy (MMD) and Wasserstein distance are symmetric: the value is the same regardless of which domain is the source and which one is the target. We emphasize the importance of asymmetric distance scores for the domain adaptation setting, once it is expected that transferring from a hard to an easy domain should be easier than transferring in the opposite direction. We cite one example in Table 1a, where the average accuracy achieved on the target domain in the P→C setting is 52.4, whereas in the C→P setting, it reaches 70.0 when using the ResNet-50 backbone.
>
> Nevertheless, to address your concern, we are currently conducting an additional comparison between our proposed PAS score and the symmetric distance metrics MMD and Proxy A-distance (as requested by reviewer YdsH). We will share our findings with the reviewers as soon as possible and include them in a revised version of our submission.
>
> Regarding the averaged prediction entropy, PAS is intended to be calculated before domain adaptation, so the model has not yet been trained on the task of interest. The entropy of the target predictions is not available, and usually, the pre-trained task (e.g., ImageNet-1k classification) is unrelated to the domain adaptation task.
>
> __W3: Failure cases__
>
> We provide in the paper the results for the benchmark ImageCLEF, for which the PAS score has a weak correlation. By the end of section 4.1, we discuss the reason for the failure and provide some examples in the Supplementary Material A.2. In short, the PAS fails in cases where a single sample contains objects from multiple classes.
>
> Besides the mentioned failure case, we could not identify in our experiments another scenario that causes a systematic failure of the PAS score.

---

> > ### Comment · Reviewer_mW8M · 2025-11-24
> >
> > I would like to appreciate author’s response to my concerns. Nevertheless, there are still more concerns remaining.
> >
> > For example, as for the 2nd weakness, PAS constructs prototypes using a pre-trained model, source data, and source labels, then computes distances to obtain the PAS score. **The process of calculating the entropy score is very similar to PAS; we can derive logits from the distance or cosine similarity between each sample and the source prototype, and then obtain the average prediction entropy score via softmax with an appropriate temperature.**

---

> ### Author Response · Authors · 2025-11-25
>
> Please see our general comment, where we share our additional experiments comparing our proposed PAS score with the symmetric distribution distance metrics MMD and $\mathcal{A}$-distance.
>
> We are updating the paper to reflect the additional experiments and modifications suggested by the reviewers.

---

### Official Review · Reviewer_j93r · 2025-10-28

**Soundness:** 4
**Presentation:** 3
**Contribution:** 3
**Rating:** 8
**Confidence:** 4

**Summary:**

This paper proposes a new method to estimate the right selection of pre-trained checkpoints and source domains for achieving highest accuracy on a particular target domain. Specifically, they rely on the ratio of the distance between a target samples and the nearest and second-nearest source prototype as a measure of transferability. Though simple, this measure is very intuitive and equally effective in the experiments. Results across various domain transfers show the string correlation between PAS score and eventual DA accuracy.

**Strengths:**

1. The paper addresses a very important problem of estimating the right choice of source domain and pre-trained model + architecture while faced with optimizing accuracy on an unlabeled target domains with significant domain shift.

2. The presented hypothesis is very simple and easily usable in wide variety of scenarios indicating the strong effectiveness of the approach.

3. The distance measure presented in Eq 3 will be small if either (i) the target sample is equidistant from two source classes (poor choice of source domain), or (ii) the two source classes are very close to each other (poor discriminability of source features), covering both the choice for source domain and source pre-trained model efficiently well.

4. The strong correlation of PAS score with various transfer settings verifies the effectiveness of proposed hypothesis.

**Weaknesses:**

1. The experiments only consider very basic transfer datasets which are relatively small scale for modern DA. Larger-scale datasets like DomainNet [1] and GeoNet [2] are not covered limiting the insights we can draw from this work for practical use-case settings.

2. There are no sufficient ablations provided in the experiments regarding other potential measures of transferability. For instance, one measure could just be the average distance between the feature representations of source and target domains in the feature space, and improvements over this simple baseline setting could be helpful.


[1] Peng, Xingchao, et al. "Moment matching for multi-source domain adaptation." Proceedings of the IEEE/CVF international conference on computer vision. 2019.
[2] Kalluri, Tarun, Wangdong Xu, and Manmohan Chandraker. "Geonet: Benchmarking unsupervised adaptation across geographies." Proceedings of the IEEE/CVF Conference on Computer Vision and Pattern Recognition. 2023.

**Questions:**

1. The experiments currently do not include the choice of which source domain is most suitable for a given target domain, although this is presented as one of the motivations in the introduction. This ablation will be helpful to have in the paper.

2. It is not completely clear how the centroid of the features are calculated in Eq1. Isn't centroid just the average of the vectors? Why is there an additional normalization in the denominator?

3. As it is mentioned that this method can also be used to choose which source domain matches a particular target domain, it would also be helpful to show why we can't just use _all_ the source domains instead of specifically choosing the best.

4. It is not clear if this method will be applicable to open-world domain adaptation methods and what modifications need to be performed to adapt this to a more general setting.

---

> ### Author Response · Authors · 2025-11-19
>
> Thank you for your thoughtful feedback on our work. We hope to address the concerns that have been raised.
>
> __W1: Larger-scale datasets.__
>
> Since GeoNet is a more recent benchmark, it has been considered by fewer domain adaptation methods in the literature. Our focus is to collect a larger and diverse set of domain adaptation methods to demonstrate that our results are consistent and that our PAS score can be applied regardless of the chosen adaptation method. Nevertheless, we are preparing additional experiments on the benchmark DomainNet and will share them soon with the reviewers. We hope that the experiments on DomainNet illustrate the applicability of PAS on larger datasets.
>
> __W2: Other transferability measures.__
>
> As suggested by the reviewers YdsH and mW8M, we are conducting additional experiments to compare our proposed PAS score with symmetric distance metrics Proxy A-distance and Maximum Mean Discrepancy (whose reasoning is related to the centroid of representations). We will share the results of the additional experiments with the reviewers soon.
>
> __Q1: The choice of a source domain for a given target domain.__
>
> We address the choice of source domains in Table 1. For each target domain, we highlight in blue the source domain with the higher PAS score, which is the recommended choice for adaptation to the target domain. We also highlight the best average final target accuracy for each target dataset. In our experiments, the source domain with the higher PAS score leads to the highest target accuracy in 44 of the 55 (80%) considered scenarios. We give more details of this experiment in section 4.1.
>
> __Q2: Additional normalization of centroids.__
>
> The arithmetic mean of a cluster usually does not have unit Euclidean length. Our work follows the procedure adopted by the popular Spherical K-means algorithm [1] for clustering high-dimensional data. The centroid is normalized so it represents the vector that, on average, has the highest cosine similarity to all the samples in the cluster. This property is derived from the Cauchy-Schwarz inequality $\sum_{x \in S^S_c} x^T v \leq \sum_{x \in S^S_c} x^T \mu_c$, that holds if $\mu_c$ is normalized for any unit vector $v$ and class $c$.
>
> [1] Dhillon, Inderjit S., and Dharmendra S. Modha. "Concept decompositions for large sparse text data using clustering." Machine learning 42.1 (2001): 143-175.
>
> __Q3: Adaptation with all the source domains.__
>
> Our work focuses on the single-source domain adaptation scenario, but we believe that the selection of multiple source domains would be an interesting evolution for future researchers. We mention this improvement as future work in Section 5. Despite believing in the benefits of using multiple source domains for adaptation, we do not encourage the use of all available domains without any previous selection, as we explain in the introduction of our work. We argue that noisy and unrelated domains can potentially lead to negative transferring.
>
> __Q4: Application to open-world domain adaptation methods.__
>
> Could you please elaborate on what would be considered open-world domain adaptation methods?
>
> Regarding the label distribution, our work focuses on the closed-set domain adaptation setting, where the label distributions are identical across the source and target domains. We acknowledge the importance of the open-set domain adaptation setting, where the label distributions can differ across domains; however, we leave this scenario for future work. We will make our scope more explicit in the final version of our paper and cite the open-work domain adaptation setting as a proposal for future work.

---

> ### Author Response · Authors · 2025-11-25
>
> Following are our experiments for the benchmark *DomainNet* and pre-trained backbone *Resnet-101*.
>
> Please refer to our general comment for an overview of all our experiments.
>
> We are updating the paper to reflect the additional experiments and modifications suggested by the reviewers.
>
> |    | P→C| R→C | S→C | C→P | R→P | S→P | C→R | P→R | S→R | C→S | P→S | R→S |Correlation with acc. (Pearson / Spearman)   |
> |---|-------|-------|--------|-------|--------|-------|--------|-------|--------|-------|--------|-------|:-----------------------------------------------------:|
> |  Acc. (avg.) | 45.5 | 53.7 | __56.7__ | 39.4 | __52.2__ | 45.8 | 55.9 | __58.1__ | 55.3 | __44.8__ | 40.7 | 41.0 |
> | MMD (neg.) | -0.113 | -0.158 | __-0.079__ | -0.113 | __-0.075__ | -0.108 | -0.158 | __-0.075__ | -0.173 | __-0.079__ | -0.108 | -0.173 | 0.04 / 0.20 |
> | $\mathcal{A}$-distance (neg.) | -1.789 | -1.73 | __-1.638__ | -1.789 | __-1.656__ | -1.777 | -1.73 | __-1.656__ | -1.821 | __-1.638__ | -1.777 | -1.821 | 0.50 / 0.45 |
> | __PAS (our)__ | 0.108 | __0.145__ | 0.088 | 0.08 | __0.159__ | 0.083 | 0.128 | __0.184__ | 0.107 | 0.088 | 0.098 | __0.114__ | __0.58__ / __0.53__ |
> | Oracle* | -0.109 | -0.124 | -0.042 | -0.06 | -0.024 | -0.04 | 0.037 | 0.092 | 0.031 | -0.087 | -0.11 | -0.156 | 0.70 / 0.67 |

---

### Official Review · Reviewer_YdsH · 2025-10-30

**Soundness:** 3
**Presentation:** 2
**Contribution:** 2
**Rating:** 2
**Confidence:** 4

**Summary:**

The paper proposes a simple score to predict how well a pre-trained model will perform when adapted to a new domain. The score is based on distances to class centroids. Experiments show that the score correlates with target domain performance.

**Strengths:**

* The proposed method is clearly explained, and should be easy to implement.
* Experiments show a good correlation between PAS score and target accuracy.
* The experiments use standard datasets for domain adaptation.
* I did not find any spelling or grammar errors.

**Weaknesses:**

* There is no comparison to related work.
  * The paper claims that "To the best of our knowledge, this is the first proposal for transferability estimation for the domain adaptation setting." But there have actually been investigations into this before. In particular there is the Proxy A-distance (Shai Ben-David, John Blitzer, Koby Crammer, and Fernando Pereira; Analysis of representations for domain adaptation; NIPS 2006), and there should be a comparison with this A-distance.
  * The paper also makes no comparison to methods that first perform domain adaptation and then try to estimate performance, because they claim that these methods are too costly. However, several domain adaptation methods only require a single evaluation of the source model, followed by a relatively cheap adaptation procedure. This should be comparable in cost to the proposed PAS score. An experimental comparison with these methods must be made, which can include a comparison of the computational cost.
* Figure 3 combines multiple datasets, to show a correlation between PAS score and target accuracy. But there are large differences between the different datasets, while the within dataset correlation is perhaps less clear. It is not appropriate to compute a single correlation coefficient for the combined datasets.

**Questions:**

* What correlation coefficient is used in the paper? Spearman's rank correlation would be more appropriate than Pearson's correlation, because the relation between PAS scores and target accuracy might be nonlinear, and the order is probably most important.
* Why is the PAS score using (d2-d1)/d2? I don't see any motivation for this choice.
* Table 1: What domain adaptation method is this using? Or are you simply applying model on target?
* In my opinion, the pseudo code in Algorithm 1 doesn't add much to the paper, since the explanation in Eq (1)-(3) is much more clear, and the pseudo code does not match how the algorithm would be implemented in a modern framework. In addition, the placement in the paper makes the surrounding text harder to read.

---

> ### Author Response · Authors · 2025-11-17
>
> Thank you for your thoughtful feedback on our work. We hope to address the concerns that have been raised.
>
> __W1: Proxy A-distance.__
>
> The Proxy A-distance is indeed a simple and intuitive measure of the distance between two domains, as well as other similar metrics we cite in the paper, such as Maximum Mean Discrepancy (MMD). However, as we mention in the introduction, the problem with such metrics is that they are symmetric, i.e., the distance is the same regardless of which domain is the source and which is the target. We argue that the asymmetry is important for domain adaptation because transferring from an easy to a hard domain is expected to be harder than transferring from a hard to an easy one. In Table 1a, for example, for the ResNet-50 backbone, the resulting average accuracy on the target domain in the setting P→C is 52.4, while in the setting C→P is 70.0.
>
> Nevertheless, to address your concern, we are currently conducting an additional comparison between our proposed PAS score and the symmetric distance metrics Proxy A-distance and MMD (as requested by reviewer mW8M). We will share our findings with the reviewers as soon as possible and include them in a revised version of our submission.
>
> __W2: Methods that first perform domain adaptation and then try to estimate performance.__
>
> You mention that we should compare against domain adaptation methods that require only a single evaluation of the source model. We think we do not completely understand the request. In the unsupervised domain adaptation setting, one is given an unlabeled target domain, different options of pre-trained models (e.g., ResNet-50, ViT) and one or more options of source domains. Performing domain adaptation typically requires finetuning the pre-trained model using gradient descent to learn the new classification task (probably different from the pre-trained task) while reducing the shift between domains. Also, when training a domain adaptation method, it is desirable to perform hyperparameter tuning, significantly increasing the final number of models to train and compare. Could you please clarify and give an example of methods that we should compare?
>
> __W3: Correlation per dataset.__
>
> Due to the lack of space, we show in the Supplementary Material A.1 the results for each individual dataset, including the correlation per domain adaptation method. We would also be happy to include in the final version of the paper figures showing the correlation for each domain. In summary, the Pearson correlation for the benchmarks Office-Home, Office-31, ImageCLEF and Office+Caltech10 is __0.76__, __0.63__, __0.44__ and __0.73__, respectively. We address in the main paper the reason for the low correlation for the ImageCLEF benchmark.
>
> __Q1: Spearman's rank correlation.__
>
> Thanks for the suggestion. We use the Pearson correlation in the main paper, but we would be happy to add the Spearman’s rank correlation as well. The overall Spearman’s rank correlation among all datasets is __0.81__ (the Pearson correlation is __0.74__). The individual correlation for the benchmarks Office-Home, Office-31, ImageCLEF and Office+Caltech10 is __0.79__, __0.83__, __0.51__ and __0.6__, respectively.
>
> __Q2: Motivation for the PAS score.__
>
> The PAS score is inspired by the Silhouette Score, which measures the intra-class cohesion and inter-class separation of clusters. The Silhouette Score is computed using __(d2-d1)/max{d1, d2}__, where d1 is the mean distance from the sample to other samples in the same class and d2 is the distance to the nearest centroid of a different class. We assume that samples from the same class are represented close together, regardless of the domain, as they likely have more characteristics in common. As a result, we expect the closest source class cluster to be the true class of the target sample. The closer the target sample is to the source class centroid (and far from other source centroids), the more characteristics the sample has in common with that class prototype. Assuming d1 as the distance to the closer source class and d2 the distance to the second closer source class result in our proposed PAS score calculated as __(d2-d1)/d2__. We provide more details by the end of section 3.3 of the paper and discuss other design choices in section 4.4.
>
> __Q3: Domain adaptation method for table 1.__
>
> Table 1 is a summary of the target accuracy of many different domain adaptation methods proposed in the literature in the past years. Due to the lack of space, we report in Table 1 the average target accuracy of the different methods, but we also provide in the Supplementary Material A.1 the complete tables showing the individual results. Nevertheless, in Figure 3, each box represents the distribution of the accuracy of the different domain adaptation methods we considered for a better overview.
>
> __Q4: Pseudo-code.__
>
> We appreciate the feedback. We propose to remove the pseudo-code from the final version for better readability.

---

> ### Author Response · Authors · 2025-11-25
>
> Please see in our general comment the additional experiments comparing our PAS score with symmetric distribution distance metrics. As requested, we included the baselines MMD and $\mathcal{A}$-distance in our experiments and reported the correlation between each score and the average target accuracy after domain adaptation. We followed your suggestion and reported both the Pearson correlation and the Spearman’s rank correlation, as well as the correlation for each benchmark.
>
> We are updating the paper to reflect the additional experiments and modifications suggested by the reviewers.

---

### Author Response · Authors · 2025-11-25

We are happy to share the additional experiments comparing our proposed score PAS with the symmetric measures of distance between domains MMD and $\mathcal{A}$-distance.

As requested by reviewer *YdsH*, we report both the Pearson (left) and the Spearman’s rank correlation (right). Also, we added experiments for the benchmark *DomainNet*, as requested by reviewer *j93r*.

|                      | Office-Home | Office-31 | ImageCLEF | Office+Caltech-10 | DomainNet | Total |
|----------------|----------------|------------|---------------|----------------------|---------------|------|
| MMD | 0.55 / 0.56 | 0.45 / 0.72 | -0.14 / -0.12 | 0.25 / 0.11 |  0.04 / 0.20 | 0.14 / 0.56 |
| $\mathcal{A}$-distance | 0.33 / 0.55 | 0.26 / 0.36 | -0.13 / -0.12 | 0.41 / 0.14 | 0.50 / 0.45 | 0.01 / 0.58 |
| __PAS (our)__ | __0.76__ / __0.79__ | __0.63__ / __0.83__ | __0.44__ / __0.51__ | __0.73__ / __0.65__ | __0.58__ / __0.53__ | __0.79__ / __0.78__ |
| Oracle* | 0.89 / 0.90 | 0.71 / 0.92 | 0.78 / 0.79 | 0.76 / 0.60 | 0.70 / 0.67 | 0.88 / 0.87 |

Due to the quadratic nature of MMD, we restrict its computation to a maximum of 10,000 randomly selected samples per domain for the *DomainNet* benchmark. The $\mathcal{A}$-distance is computed using C-Support Vector Classification.

---

> ### Comment · Reviewer_mW8M · 2025-11-26
>
> I appreciate the authors' significant efforts in conducting additional experiments. **However, as shown in Table 1 (a)-(e) in the manuscript, the PAS metric does not clearly outperform MMD and A-distance.**
>
> For instance, on the most challenging dataset, DomainNet (Table 1(e)), PAS fails to identify the appropriate source model, whereas both MMD and A-distance succeed.
>
> Furthermore, this table indicates that "correlation with accuracy" is not a reliable measure for evaluating the effectiveness of different metrics in model selection.

---

> ### Comment · Reviewer_YdsH · 2025-11-27
>
> For DomainNet, the computed correlation coefficients in table 1(e) of the updated paper do not match the rest of the table. If I compute them I find
>
> |          | MMD       | A-dist    | PAS       | Oracle   |
> |----------|-----------|-----------|-----------|----------|
> |Pearson:  | 0.038988  | 0.500432  | 0.586235  | 0.704019 |
> |Spearman: | 0.197891  | 0.452322  | 0.507882  | 0.671329 |

---

> > ### Author Response · Authors · 2025-11-27
> >
> > Thank you for pointing out the error. We had incorrectly correlated the scores with PAS (instead of the accuracy). We double-checked the remaining values and they are correct. We are going to update our previous comments.

---

### Author Response · Authors · 2025-11-26

We uploaded a new version of the paper with the following modifications:

- Included the baselines MMD and $\mathcal{A}$-distance.
- Included the benchmark DomainNet.
- Correlation reported for each benchmark.
- Reported the Pearson and the Spearman's rank correlation.
- Pseudo-code removed for better readability.

---

### Author Response · Authors · 2025-12-03
**Summary to the Area Chair**

Following is a summary of our discussion with the reviewers, along with the improvements we implemented in response to their feedback during the rebuttal phase.

__Comparison with other baselines__

The main weakness raised by all 3 reviewers is the lack of comparison with other baselines. To address the concerns, we presented experiments comparing our PAS score with the symmetric distance metrics __$\mathcal{A}$-distance__ and __MMD__ (both proposed by the reviewers). We showed that the correlation between our score and the final target accuracy is *significantly* higher than the baselines across all the considered benchmarks, from those originally considered in the work to those suggested by the reviewers.

__Experiments on a larger benchmark__

As solicited by reviewer *j93r* (rating 8, confidence 4), we added experiments for the bigger and more challenging benchmark __DomainNet__. We reported results for the ResNet-101 pre-trained backbone, showing higher overall correlation than the baselines. The results raised the criticism of reviewer *mW8M* (rating 4, confidence 4) regarding the reliability of using the overall correlation of the scores with the final target accuracy. The reviewer noted that our PAS score selected the source domain with the highest accuracy in only 2 of the 4 scenarios for this benchmark. To address the reviewer's criticism, we present a more complete set of experiments for this benchmark, considering 3 additional pre-trained backbones. In these extended experiments, our PAS score outperforms the baselines in both the correlation with the target accuracy and the number of correctly selected source domains.

Regarding the use of the overall correlation, we agree with the criticism of reviewer *mW8M* and present the correlation *per target domain*. This is the most relevant correlation for users in real-world scenarios. Given a target domain and many options of source domains and pre-trained models, our PAS score has a strong correlation with the final target accuracy. We compare our PAS score with the __$\mathcal{A}$-distance__ and __MMD__ baselines proposed by the reviewers, and with an __Oracle__, which is similar by our score, but considers the target labels (unavailable in real scenarios). Please note that these are new experiments, not previously shared with the reviewers.

Spearman’s rank correlation per target domain:

|                | **Office-Home** |          |          |          | **Office-31** |          |          | **ImageCLEF** |          |          | **DomainNet** |          |          |          |
|----------------|-----------------|----------|----------|----------|---------------|----------|----------|---------------|----------|----------|---------------|----------|----------|----------|
|                |      **A**      |   **C**  |   **P**  |   **R**  |     **A**     |   **D**  |   **W**  |     **C**     |   **I**  |   **P**  |     **C**     |   **P**  |   **R**  |   **S**  |
|     **MMD**    |       0.26      |   0.21   |   0.29   |   0.21   |     -0.15     |   0.60   |    0.36    |      0.49     |   -0.22  |   0.60   |     -0.42     |   0.20   |   0.47   |   -0.42  |
| **A-distance** |      -0.13      |   -0.43  |   -0.09  |   -0.26  |     -0.31     |   0.28   |   0.07   |      0.38     |   0.05   |   0.60   |     -0.04     |   0.17   |   0.30   |   -0.32  |
|     **PAS (our)**    |     **0.70**    | **0.78** | **0.74** | **0.75** |    **0.81**   | **0.81** | **0.75** |    **0.76**   | **0.66** | **0.83** |    **0.67**   | **0.76** | **0.71** | **0.35** |
|   **Oracle**   |       0.85      |   0.90   |   0.81   |   0.87   |      0.88     |   0.90   |   0.78   |      0.76     |   0.66   |   0.94   |      0.50     |   0.62   |   0.72   |   0.33   |

Overall Spearman’s rank correlation per benchmark (updated after additional experiments):

|                | **Office-Home** | **Office-31** | **ImageCLEF** | **DomainNet** | **Total** |
|----------------|-----------------|---------------|---------------|---------------|-----------|
|     **MMD**    |       0.51      |      0.53     |     -0.08     |     -0.03     |    0.37   |
| **A-distance** |       0.17      |      0.35     |     -0.07     |      0.06     |   -0.16   |
|     **PAS (our)**    |     **0.81**    |    **0.78**   |    **0.60**   |    **0.56**   |  **0.88** |
|   **Oracle**   |       0.90      |      0.86     |      0.85     |      0.21     |    0.91   |

We corrected the Spearman’s rank correlation that we had computed incorrectly in our previous comments.

(continue)

---

> ### Author Response · Authors · 2025-12-03
>
> __Other improvements__
>
> - We addressed other improvements suggested by the reviewers:
> - Reported the correlation per benchmark. (reviewer *YdsH*, rating 2, confidence 4)
> - Reported the Spearman's rank correlation. (reviewer *YdsH*, rating 2, confidence 4)
> - Removed the pseudo-code for better readability. (reviewer *YdsH*, rating 2, confidence 4)
> - Improved the explanation of the computation of the clusters’ centroid. (reviewer j93r, rating 8, confidence 4).
>
> We responded to the remaining questions raised by the reviewers and asked for additional explanations for the few questions we do not fully understand. Unfortunately, we did not have feedback for these questions.
>
> When performing the additional experiments, we noted that only a few works reported results for the benchmark __Office+Caltech-10__. The limited number of domain adaptation methods and pre-trained backbones for this benchmark makes the comparison less reliable. For this reason, we decided not to report the experiments for this benchmark in our work anymore. This decision should have a little impact, since most of this benchmark is a subset of __Office-31__, whose results we report in our work.
>
> __Conclusion__
>
> We addressed all the reviewers' criticisms. Our work was improved with additional experiments for 2 new baselines and 1 benchmark. We show that our PAS score consistently outperforms the baselines in all experiments by a significant factor. With no remaining weaknesses or questions to address, we believe we have met the reviewer’s expectations, and we kindly request that our score be increased.

---

### Meta-Review · Area_Chair_KXEG · 2026-01-06

**Summary:**

The paper initially received mostly negative scores: 8, 4, 2. The main concerns include: (1) comparisons with other metrics; (2) experiments on larger datasets; (3) additional experimental results; and (4) further clarification of technical details. The AC has carefully read the reviews and the rebuttal, and finds that the authors have largely addressed these concerns.

Specifically, for the concerns raised by Reviewer YdsH and Reviewer j93r, the authors have provided comparisons with additional baselines and clearer explanations of technical details.

For the concerns raised by Reviewer mW8M, the authors have also addressed issues related to overstatement, comparisons with additional baselines, and failure-case analysis.

AC comment: The AC also notes that a highly related work is missing from the paper: [A], which similarly aims to identify the best source model for target domains in an unsupervised manner. Although [A] mainly focuses on retrieval, the AC strongly encourages the authors to compare with and discuss this work in the final version.

[A] Ranking Models in Unlabeled New Environments. ICCV 2023.

Given these considerations, the AC believes the main concerns have been addressed, and the reviewers are likely to revise their scores to be positive (e.g., 8, 6, 6). The AC therefore recommends acceptance, and encourages the authors to incorporate all comments from the reviewers and the AC in the final version.

**Reviewer Concerns:**

Solved Concerns:

Reviewer YdsH: comparisons with additional baselines and clearer explanations of technical details.

Reviewer j93r: comparisons with additional baselines and clearer explanations of technical details.

Reviewer mW8M: issues related to overstatement, comparisons with additional baselines, and failure-case analysis.

Remaining Concerns:

Reviewer j93r: Results using all source domains and experiments on open-world domain adaptation. The authors consider these two points as future work.

**Reviewer Scores:**

Reviewer YdsH may change the score from 2 to 6 as the main concerns were solved.

Reviewer j93r may keep the score of 8 as the main concern were solved but two minor concerns were not solved.

Reviewer mW8M may change the score from 4 to 6 as the main concerns were solved.

---

### Decision · Program_Chairs · 2026-01-26

Accept (Poster)